# A novel mitochondrial Kv1.3–caveolin axis controls cell survival and apoptosis

Jesusa Capera[1†§], Mireia Pérez-Verdaguer[1†#], Roberta Peruzzo[2‡], María Navarro-Pérez[1‡], Juan Martínez-Pinna[3], Armando Alberola-Die[3], Andrés Morales[3], Luigi Leanza[2], Ildiko Szabó[2*], Antonio Felipe[1*]

[1]Molecular Physiology Laboratory, Dpt. de Bioquímica i Biomedicina Molecular, Institut de Biomedicina (IBUB), Universitat de Barcelona, Barcelona, Spain; [2]Department of Biology, University of Padova, Padova, Italy; [3]Dept de Fisiología, Genética y Microbiología, Universidad de Alicante, Alicante, Spain

*For correspondence:
ildi@mail.bio.unipd.it (IS);
afelipe@ub.edu (AF)

[†]These authors contributed equally to this work
[‡]These authors also contributed equally to this work

Present address: [§]Kennedy Institute of Rheumatology, University of Oxford, Oxford, United Kingdom; [#]Department of Cell Biology, University of Pittsburgh School of Medicine, Pittsburgh, United States

Competing interests: The authors declare that no competing interests exist.

**Abstract** The voltage-gated potassium channel Kv1.3 plays an apparent dual physiological role by participating in activation and proliferation of leukocytes as well as promoting apoptosis in several types of tumor cells. Therefore, Kv1.3 is considered a potential pharmacological target for immunodeficiency and cancer. Different cellular locations of Kv1.3, at the plasma membrane or the mitochondria, could be responsible for such duality. While plasma membrane Kv1.3 facilitates proliferation, the mitochondrial channel modulates apoptotic signaling. Several molecular determinants of Kv1.3 drive the channel to the cell surface, but no information is available about its mitochondrial targeting. Caveolins, which are able to modulate cell survival, participate in the plasma membrane targeting of Kv1.3. The channel, via a caveolin-binding domain (CBD), associates with caveolin 1 (Cav1), which localizes Kv1.3 to lipid raft membrane microdomains. The aim of our study was to understand the role of such interactions not only for channel targeting but also for cell survival in mammalian cells. By using a caveolin association-deficient channel (Kv1.3 CBD*less*), we demonstrate here that while the Kv1.3–Cav1 interaction is responsible for the channel localization in the plasma membrane, a lack of such interaction accumulates Kv1.3 in the mitochondria. Kv1.3 CBD*less* severely affects mitochondrial physiology and cell survival, indicating that a functional link of Kv1.3 with Cav1 within the mitochondria modulates the pro-apoptotic effects of the channel. Therefore, the balance exerted by these two complementary mechanisms fine-tune the physiological role of Kv1.3 during cell survival or apoptosis. Our data highlight an unexpected role for the mitochondrial caveolin–Kv1.3 axis during cell survival and apoptosis.

## Introduction

The voltage-gated potassium channel Kv1.3 is present at the plasma membrane of different cell types, mostly neurons and leukocytes (*Cahalan and Chandy, 2009*; *Martínez-Mármol et al., 2016*; *Solé et al., 2016*). Kv1.3 participates in cell proliferation, activation, and apoptosis (*Pérez-Verdaguer et al., 2016b*). Thus, altered expression of the channel is linked to different pathologies such as autoimmune diseases and cancer (*Rus et al., 2005*; *Vallejo-Gracia et al., 2013*; *Serrano-Novillo et al., 2019*; *Szabo et al., 2021*). Kv1.3 is efficiently expressed on the cell surface, which depends on multiple forward trafficking signatures located at the C-terminal domain of the channel (*Martínez-Mármol et al., 2013*). Moreover, different ancillary interactions modulate the trafficking of Kv1.3 (*Capera et al., 2019*). For example, the Kv1.5 channel and the regulatory KCNE4 subunit retain Kv1.3 at the endoplasmic reticulum (ER), negatively modulating the channel surface expression (*Vicente et al., 2006*; *Vicente et al., 2008*; *Solé et al., 2009*). In addition, the scaffolding protein caveolin 1 (Cav1) controls Kv1.3 spatial localization in raft microdomains, which is important for signaling and cell physiology (*Pérez-Verdaguer et al., 2016a*; *Pérez-Verdaguer et al., 2018*).

Therefore, by controlling Kv1.3 surface expression and localization, oligomeric associations fine-tune physiological events.

Caveolin (Cav) is the main structural component of caveolae, a specialized form of membrane lipid raft with a characteristic omega-shaped structure. Caveolae are abundant at the plasma membrane of highly differentiated cells, such as adipocytes, pneumocytes, and muscle cells, but they are not abundant in central neurons and lymphocytes. Cav1, the main isoform in non-muscle tissues, forms large oligomers with high affinity for cholesterol and sphingolipids. In addition, Cav1 can recruit different proteins into caveolar and non-caveolar rafts through its Cav scaffolding domain (CSD). Therefore, rafts, acting as signaling platforms, initiate signaling pathways, participate in vesicular transport, and contribute to cholesterol homeostasis (*Simons and Ikonen, 1997*; *Razani et al., 2002*; *Parton et al., 2006*). In addition, these microdomains are essential for immunological synapses during the immune response (*Rao et al., 2004*), as well as for the insulin modulation of adipocyte physiology (*Pérez-Verdaguer et al., 2018*).

In this context, the role of Cav1 in cancer progression raises an intense debate. Cav1 acts either as a tumor suppressor or as an oncogene, depending on the cancer type and the clinical stage of the disease. For instance, low Cav1 expression is associated with low survival in stromal breast cancer cells, whereas high expression of Cav1 indicates poor prognosis in invasive breast cancer cells (*Qian et al., 2011*). Thus, Cav1 regulates different oncogenic properties, such as neoplastic transformation, apoptosis resistance, migration, invasiveness, and angiogenesis (*Quest et al., 2008*; *Qian et al., 2019*), likely depending on the cell type and/or Cav1 interactions with specific partners. Reciprocal regulation between Cav1 and the cell oxidative state regulates cell survival and stress-dependent responses (*Wang et al., 2017*). In addition, Cav1 also plays non-caveolar functions (*Volonte et al., 2016*). Cav1 localizes not only to the plasma membrane, but also to mitochondria, where it participates in the regulation of cell bioenergetics and apoptosis and, consequently, in cancer progression (*Nwosu et al., 2016*). Cav1 upregulation promotes apoptosis resistance and provides a metabolic advantage to cancerous cells (*Wang et al., 2017*). In fact, Cav1 inhibits Bax-dependent cell death, helping cancer cells to escape chemotherapy (*Zou et al., 2012*; *Shiroto et al., 2014*). In contrast, Cav1 knockdown has been reported to cause hyperpolarization of the inner mitochondrial membrane (IMM) potential ($\Delta\psi_m$) and to alter the lipid composition of the IMM, leading cells to apoptosis (*Bosch et al., 2011*).

Kv1.3, which is present in lipid rafts (*Bock et al., 2003*) and associates with Cav1, participates in apoptosis and chemotherapy resistance through its mitochondrial localization (*Szabó et al., 2008*; *Leanza et al., 2012*). Mitochondrial Kv1.3 (mitoKv1.3) mediates the pro-apoptotic effects of Bax. Bax blocks Kv1.3 causing the hyperpolarization of the IMM and a subsequent reactive oxygen species (ROS) production. These events lead to the opening of the permeability transition pore (PTP) and to consequent IMM depolarization, which is followed by the release of cytochrome c and the triggering of the intrinsic apoptotic cascade. Sustained PTP opening leads to the loss of mitochondrial integrity and respiration and induces swelling (*Szabó et al., 2008*; *Szabò et al., 2011*). The effect of Bax, which is often downregulated in cancer cells, can be mimicked by mitochondria-targeted Kv1.3 inhibitors (*Leanza et al., 2012*). For this reason, mitoKv1.3 has become a potential target for chemotherapy and a solution for overcoming chemotherapeutic resistance (*Leanza et al., 2012*; *Leanza et al., 2017*; *Szabo et al., 2021*).

Because the Kv1.3 and Cav1 interaction in lipid rafts has an enormous influence on cell physiology (*Vicente et al., 2006*; *Pérez-Verdaguer et al., 2016a*; *Pérez-Verdaguer et al., 2018*), we analyzed the functional link between Kv1.3 and Cav1 in the regulation of intrinsic apoptosis. We demonstrated that the interaction of Kv1.3 with Cav1 is important for the plasma membrane targeting of the channel. On the other hand, once in the mitochondria, the Kv1.3–caveolin axis functions as an anti-apoptotic mechanism protecting the cells from Kv1.3-mediated cell death. Our data increases the understanding of the heterogeneity of cancer by consolidating the roles of Kv1.3 and Cav1 as targets for anti-cancer therapies.

## Results

### Interaction with caveolin-1 governs the spatial localization of Kv1.3

We have previously described that the interaction between Kv1.3 and Cav1 results in the localization of the channel to lipid rafts and caveolae (*Pérez-Verdaguer et al., 2016a*; *Pérez-Verdaguer et al., 2018*). Kv1.3 contains a caveolin-binding domain (CBD) situated on the N-terminal end of the channel. Molecular simulations of Kv1.3 identified this CBD as an α-helix in an exposed orientation of the

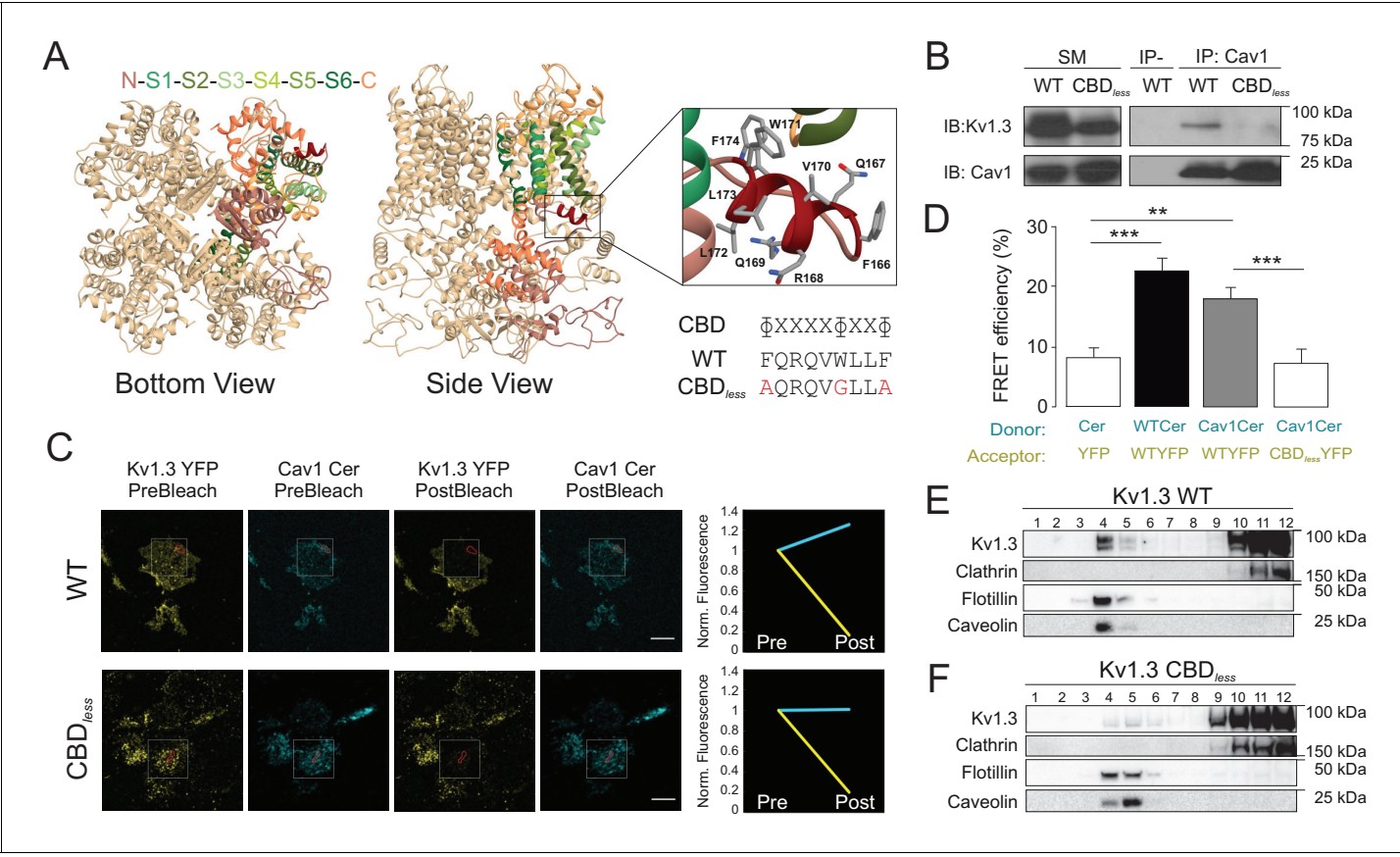

**Figure 1.** The caveolin-binding domain (CBD) of Kv1.3 mediates the interaction with Cav1 targeting the channel to lipid raft microdomains. (**A**) Ribbon representation of a Kv1.3 tetramer. For clarity, the transmembrane domain structures are highlighted with different colors in one monomer, with the CBD in red. Both the cytoplasmic (bottom view) and the side (side view) planes are shown. Note the exposed orientation of the CBD at the proximal cytoplasmic N-terminal domain. A zoomed in image is provided for detail. Aminoacids are identified with letters and positions. Lateral chains are colored by element (C, gray; N, blue; O, red). The consensus sequence of the CBD is provided. The amino acid sequence shows the CBD of wild type (WT) Kv1.3. The Kv1.3 CBD mutant (CBD*less*) contains amino acid substitutions (in red) to abrogate the CBD. (**B**) Kv1.3–Cav1 coimmunoprecipitation assay. HEK 293 Cav⁻ cells were cotransfected with Cav1 and Kv1.3YFP WT or Kv1.3YFP CBD*less*. Total cell lysates were immunoprecipitated with Cav1 (IP: Cav1). IP−, absence of Cav1 antibody. SM, starting materials. Samples were immunoblotted (IB) against Cav1 or Kv1.3. (**C**) Representative images from a Förster resonance energy transfer (FRET) experiment on cell unroofing preparations (CUPs). HEK 293 Cav⁻ cells were cotransfected with Kv1.3YFP WT+Cav1 Cerulean (Cav1 Cer) and Kv1.3YFP CBD*less*+Cav1 Cer. From left to right: acceptor (Kv1.3 YFP) and donor (Cav1 Cer) prebleach and postbleach images. Square insets indicate the bleached zone. Line graphs at the right show changes in donor (cerulean) and acceptor (yellow) fluorescence after bleaching. (**D**) FRET efficiency (%). Values are the mean ± SE (n > 25). **p<0.01, ***p<0.001 (Student's t-test). YFP+Cer were used as negative control. Positive controls were Kv1.3 YFP WT+Kv1.3 Cer WT. (**E, F**) Purification of detergent-resistant membrane fractions (lipid rafts). HEK293 cells were transfected with Kv1.3YFP WT or Kv1.3YFP CBD*less* and samples subjected to sucrose-density gradients (1–12 from low [top of tube] to high [bottom of tube] density fractions, respectively). Clathrin was used as a non-raft marker, and flotillin and caveolin as lipid raft markers.

The online version of this article includes the following figure supplement(s) for figure 1:

**Figure supplement 1.** Disruption of the CBD of Kv1.3 impairs caveolin colocalization, as well as association with the channel.

**Figure supplement 2.** Kv1.3 CBD*less* forms tetramers.

**Figure supplement 3.** Functional Kv1.3 CBD*less* channels exhibit altered biophysical properties and decreased current density at the plasma membrane.

tetrameric structure (*Figure 1A*). The sequence for the CBD (ΦxxxxΦxxΦ, where Φ is an aromatic residue and x is an unspecified amino acid) of Kv1.3 is **F**QRQV**W**LL**F**. To further study the nature of the Kv1.3–Cav1 interaction, we abrogated the CBD by replacing the aromatic residues with Ala or Gly (Kv1.3 CBD*less*, *Figure 1A*). Deletion of the CBD motif of Kv1.3 caused the loss of Kv1.3–Cav1 association, as demonstrated by the absence of co-IP and Förster resonance energy transfer (FRET) (*Figure 1B–D*). Analogous CBDs are located at the N-terminus of HCN channels, and some point mutations of this motif are sufficient to alter Cav1 binding (*Barbuti et al., 2012*). Therefore, we analyzed whether this also applied for Kv1.3 (*Figure 1—figure supplement 1*). Any substitution of aromatic residues in the CBD decreased the colocalization with Cav 1 (*Figure 1—figure supplement 1A–C*). However, similar to HCN4, only the disruption of the last pair of aromatic amino acids (W171G/F174A) greatly impaired the association with Cav 1 (*Figure 1—figure supplement 1D*).

In addition, the absence of the Cav1 interaction displaced Kv1.3 from lipid raft microdomains (*Figure 1E,F*). Although the CBD lies next to the T1 (Kv tetramerization domain), its abolition did not prevent tetramerization of the channel: (1) Kv1.3 CBD*less* formed tetramers that were observed by FRET (*Figure 1—figure supplement 2A,B*) and (2) Kv1.3 CBD*less* formed oligomeric structures observed with nondenaturing polyacrylamide gel electrophoresis (*Figure 1—figure supplement 2C*). However, the biophysical properties of the plasma membrane Kv1.3 CBD*less* were affected (*Figure 1—figure supplement 3*) as assessed in *Xenopus* oocytes. Two-electrode voltage-clamp recordings in oocytes microinjected with Kv1.3 WT or Kv1.3 CBD*less* (*Figure 1—figure supplement 3A*) indicated that Kv1.3 CBD*less* exhibited less current intensity, with a −20 mV hyperpolarized shift in the steady-state activation, compared to Kv1.3 WT (*Figure 1—figure supplement 3C–G*). In addition, the characteristic C-type inactivation of Kv1.3 was accelerated in Kv1.3 CBD*less* (*Figure 1—figure supplement 3H,I*) also exhibiting a slightly augmented cumulative inactivation (*Figure 1—figure supplement 3J–L*). Finally, both Kv1.3 channels, WT and CBD*less*, hyperpolarized the membrane potential (*Figure 1—figure supplement 3M*), but the CBD*less* mutant decreased the input resistance of the oocyte cell membrane (*Figure 1—figure supplement 3N*).

Although Kv1.3 CBD*less* was functional, an impaired Cav1 interaction altered the membrane distribution of the channel by excluding it from lipid raft structures (*Figure 1E,F*) and reduced the current density (*Figure 1—figure supplement 3*), which could be the consequence of reduced surface abundance (*Martínez-Mármol et al., 2013*). Therefore, we analyzed the targeting of Kv1.3 CBD*less* to membranes other than the plasma membrane in HEK 293 cells. The CBD motif disruption caused a notable intracellular retention (at the ER, Golgi, and mitochondria as shown below), which reduced the surface expression of the channel (*Figure 2A–D*); these data were further confirmed by biotinylation assays (*Figure 2E*). Kv1.3 usually appears as glycosylated and non-glycosylated protein forms (*Figure 2F*). Glycosylation studies, in the presence of tunicamycin, indicated an altered glycosylation for Kv1.3 CBD*less*, which mostly affected the larger glycosylated band (*Figure 2F,G*). Low levels of N-glycosylation, a reduced half-life (*Figure 2H*) and a decrease in surface expression, concomitant with a punctuated intracellular pattern for Kv1.3 CBD*less* (*Figure 2A–C*), suggested an altered maturation and stability of the channel.

N-glycosylation is an elaborated process that occurs at the ER and continues along the Golgi cisternae. To test the route of Kv1.3 CBD*less* within the cells, we disrupted ER–Golgi traffic with brefeldin A (BFA) and determined the channel distribution. As we previously described (*Martínez-Mármol et al., 2013*), BFA blocked the Kv1.3 WT trafficking and, interestingly, Kv1.3 CBD*less* exhibited a similar uniform ER distribution (*Figure 2I*).

## Kv1.3 CBD*less* accumulates in mitochondria and alters mitochondrial morphology and function

As mentioned above, the plasma membrane channel participates in cell proliferation, whereas mitoKv1.3 overexpression facilitates apoptosis (*Szabó et al., 2008*). We showed that the Cav1 interaction is essential for plasma membrane targeting of the channel and that CBD disruption triggered a punctate intracellular phenotype. Therefore, we wondered whether this scenario affected the mitochondrial localization of the channel, especially because targeting mechanisms for mitoKv1.3 are unknown. While Kv1.3 does not display a classical N-terminal mitochondria-targeting pre-sequence, membrane-permeant mitochondriotropic channel antagonists have unequivocally demonstrated a pivotal role for mitoKv1.3 in the death of primary human lymphocytic leukemia cells as well as of B16F10 melanoma cells (*Leanza et al., 2012*; *Leanza et al., 2017*). Thus, we used melanoma cells in

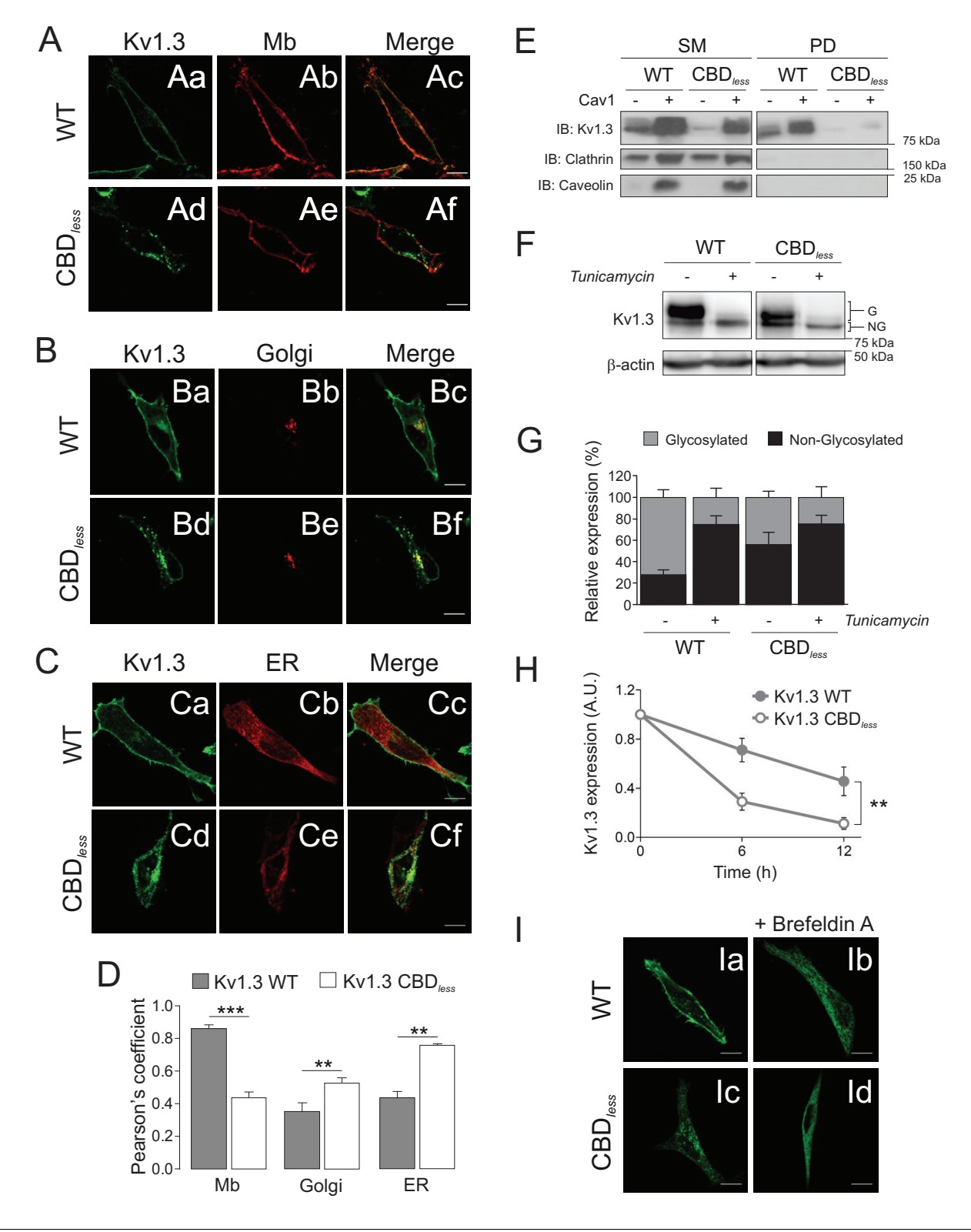

**Figure 2.** The integrity of the CBD domain is involved in the surface expression of Kv1.3. HEK 293 cells were transfected with Kv1.3YFP WT and Kv1.3YFP CBD*less*. (A–C) Representative confocal images show colocalization of Kv1.3YFP WT and Kv1.3YFP CBD*less* with (A) plasma membrane (Mb), (B) Golgi, and (C) endoplasmic reticulum (ER). Green panels, Kv1.3; red panels, subcellular marker; merge panels show colocalization in yellow. The scale bar is 10 μm. ER (pDsRed-ER) and Mb (Akt-PH-pDsRed) were used as ER and Mb markers, respectively, and were cotransfected with the channel. Golgi

*Figure 2 continued on next page*

*Figure 2 continued*

was stained with an anti-*cis*-Golgi antibody (GM130). (**D**) Colocalization analysis (Pearson's coefficient) between channel and subcellular markers. Gray bars, Kv1.3 WT. White bars, Kv1.3 CBD$_{less}$. Data are the mean ± SE (n > 30 cells) **p<0.01; ***p<0.001 vs Kv1.3 WT (Student's t-test). (**E**) Cell surface biotinylation analysis of the surface expression of Kv1.3. HEK 293 Cav$^-$ cells were cotransfected with Kv1.3YFP WT and Kv1.3YFP CBD$_{less}$ in the presence (+) or the absence (−) of Cav1. SM, starting materials. PD, pull-down (biotinylated proteins). Samples were immunoblotted (IB) for Kv1.3, clathrin (negative control), and Cav1 (caveolin). (**F**) Cells transfected with Kv1.3YFP WT or Kv1.3YFP CBD$_{less}$ were treated with (+) or without (−) 0.5 µg/ml tunicamycin for 24 hr to inhibit N-glycosylation. Total cell lysates were immunoblotted against Kv1.3 (anti-GFP) and β-actin. (**G**) Relative Kv1.3 glycosylation. The percentage of glycosylated (G) and nonglycosylated (NG) forms was calculated from data in (**F**). Data are the mean ± SE of 4 independent experiments. (**H**) Kv1.3 protein stability. Cells transfected with Kv1.3YFP WT and Kv1.3YFP CBD$_{less}$ for 24 hr were further treated for 0, 6, and 12 hr with 100 µg/ml cycloheximide. Total protein extracts were separated by SDS–PAGE and immunoblotted using Kv1.3 and β-actin antibodies. Kv1.3 expression was corrected using β-actin levels and relativized by initial values at 0 hr. A.U, arbitrary units. Data are the mean ± SE of three independent experiments. **p<0.01 (two-way ANOVA). (**I**) Representative confocal images of Kv1.3YFP WT (Ia, Ib) or Kv1.3YFP CBD$_{less}$ (Ic, Id) in the presence (Ib, Id) or in the absence (Ia, Ic) of 5 µg/ml brefeldin A (+Befeldrin A) for 4 hr. The scale bar represents 10 µm.

addition to HEK 293 cells, to decipher a general mechanism. Similar to HEK 293 cells, Kv1.3 CBD$_{less}$ exhibited less cell surface abundance than Kv1.3 WT in B16F10 melanoma cells (*Figure 3—figure supplement 1*). Interestingly, the expression of Kv1.3 CBD$_{less}$ decreased at the plasma membrane whereas augmented in the mitochondria of both HEK 293 cells (*Figure 3A–D*) and B16F10 melanoma cells (*Figure 3E,F*). Therefore, a deficient caveolin interaction, impairing surface expression, partially redirected Kv1.3 to mitochondria (*Figure 3B*), where it triggered mitochondrial network fragmentation (*Figure 4A*). Morphometric analysis of Kv1.3 CBD$_{less}$ HEK 293 cells indicated that mitochondria were smaller and more round-shaped than those observed in Kv1.3 WT cells (*Figure 4B–D*). Correlative transmission electron microscopy in both B16F10 melanoma (*Figure 4E, F*) and HEK 293 cells (*Figure 4G*) showed that only the mitochondria in cells expressing Kv1.3 CBD$_{less}$ lost cristae and became more round. In addition, mitochondrial functionality was also severely altered. Indeed, mitochondria of Kv1.3 CBD$_{less}$ cells were significantly depolarized (*Figure 5A*), and respiration was greatly impaired. The basal oxygen consumption rate (OCR) was dramatically lower in cells expressing Kv1.3 CBD$_{less}$ than in those expressing the WT channel and non-transfected cells (*Figure 5B,C*). ATP-linked respiration and the nonmitochondrial respiration, measured in the presence of oligomycin (ATPase synthase inhibitor) and antimycin (complex III blocker), respectively, were not affected. However, the maximal respiration rate, the reserve capacity and the proton leaking were significantly reduced, indicating an important loss of mitochondrial function (*Figure 5B,C*). Altogether, these data further confirmed the localization and role of both WT and CBD$_{less}$ Kv1.3 in mitochondria and pointed to their differential effects on mitochondrial physiology.

The disruption of mitochondrial morphology and the loss of mitochondrial function increased cell sensitivity toward apoptotic stimuli. HEK 293 (*Figure 6A*) and B16F10 melanoma cells (*Figure 6—figure supplement 1*) were treated with different pro-apoptotic compounds. Annexin V assays indicated that, in both cells types, Kv1.3 CBD$_{less}$ cells exhibited an elevated levels of apoptosis (*Figure 6A*, *Figure 6—figure supplement 1*), even in the absence of apoptosis-inducing agents. In this scenario, we analyzed whether the intracellular accumulation of Kv1.3, independently of CBD$_{less}$ mutation (*Figure 2D*), would affect the physiology of intracellular organelles leading to apoptosis. We took advantage of the Kv1.3 (YMVIii) mutant, which is highly ER retained (*Martínez-Mármol et al., 2013*). Concomitantly with an ER retention, Kv1.3 (YMVIii) accumulated in mitochondria (*Figure 6—figure supplement 2A,B*), which would warrant a nice control. Unlike Kv1.3 CBD$_{less}$, Kv1.3 (YMVIii) associated with Cav 1 (*Figure 6—figure supplement 2C*), but triggered no relevant apoptosis (*Figure 6—figure supplement 2D*). The effect was specific to mitochondria because there were no differences in ER-stress markers, such as GRP78, XBP1, ATF4, and the eIF2αpS51/eIF2α ratio, between Kv1.3 WT and Kv1.3 CBD$_{less}$ (*Figure 6—figure supplement 3*).

Apoptosis was also clearly visible by electron microscopy (*Figure 6D–K*). Cells that expressed Kv1.3 WT triggered no morphological changes and maintained mitochondria with sufficient cristae (*Figure 6B,C*), while Kv1.3 CBD$_{less}$ cells generated two apoptotic phenotypes: some cells were severely affected and full of apoptotic bodies, whereas others showed milder effects (*Figure 6D*). Electron micrographs with immunogold labeling further supported the accumulation of Kv1.3 CBD$_{less}$ in intracellular organelles, such as the ER, Golgi, or mitochondria (*Figure 6F–H*,

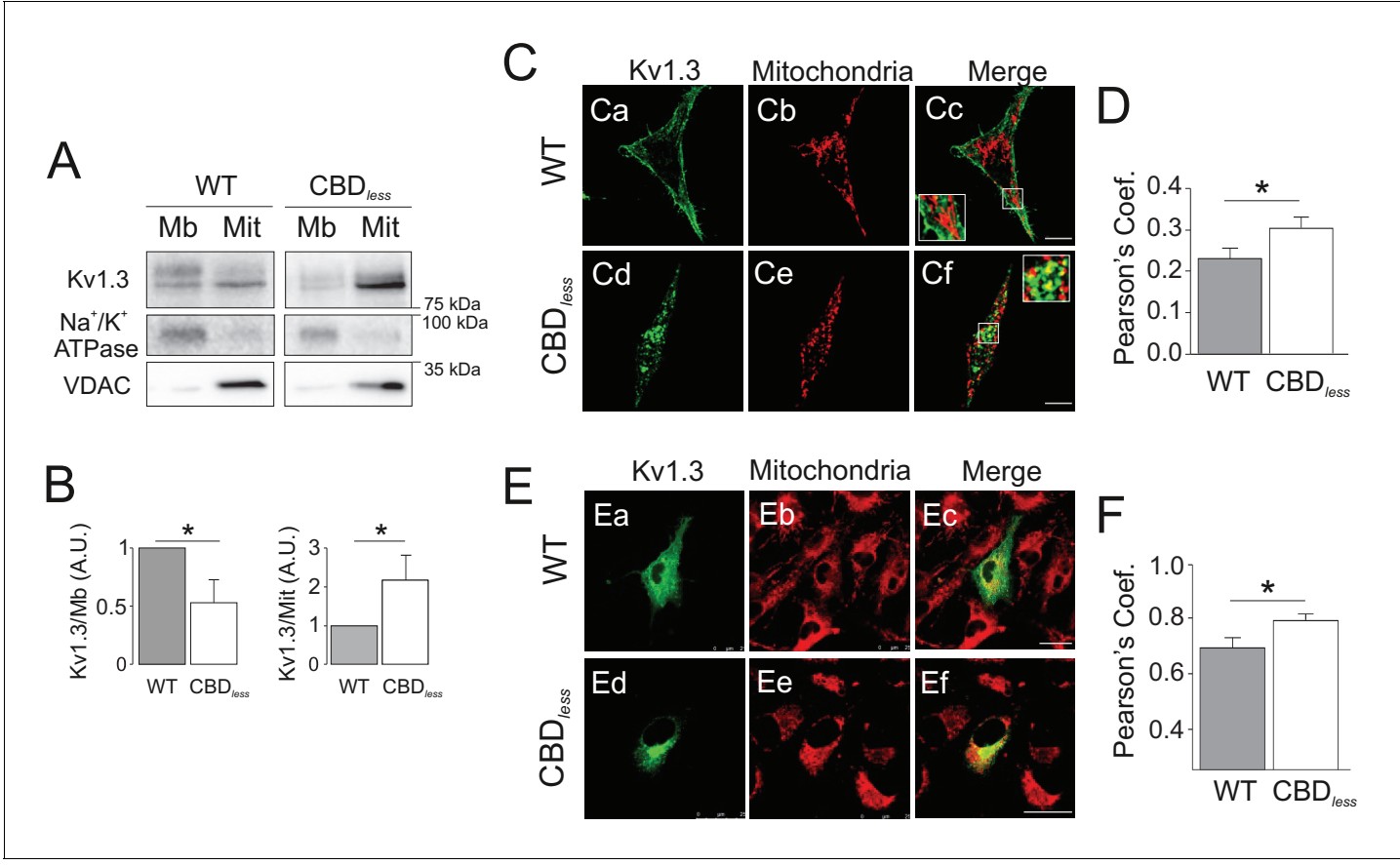

**Figure 3.** Kv1.3 CBD*less* targets to mitochondria. HEK 293 and B16F10 cells were transfected with Kv1.3YFP WT or Kv1.3YFP CBD*less*. (**A**) Subcellular fractionation isolating mitochondrial (Mit) and membrane (Mb) fractions from HEK 293 cells. Samples were immunoblotted for GFP (Kv1.3), $Na^+/K^+$ ATPase (membrane marker) or VDAC (mitochondrial marker). (**B**) Relative membrane (Mb) mitochondrial (Mit) Kv1.3 expression. Kv1.3 abundance in (**A**) was normalized to $Na^+/K^+$ ATPase (Mb) and VDAC (Mit) expression and relativized to the Kv1.3 WT. Data are the mean ± SE (n = 4). *$p<0.05$ (Student's t-test). (**C**) Representative confocal images of (Ca-Cc) Kv1.3YFP WT and (Cd-Cf) Kv1.3YFP CBD*less* (green) and mitochondria (pmitoRFP in red) from HEK 293 cells. (Cc, Cf) Merge shows colocalization in yellow. Insets magnify white squares for detail. (**D**) Quantification of colocalization was performed by Pearson's coefficient. Data are the mean ± SE (n > 30). *$p<0.05$ (Student's t-test). (**E**) Representative confocal images of (Ea–Ec) Kv1.3YFP WT and (Ed-Ef) Kv1.3YFP CBD*less* (green) and mitochondria (mitotracker in red) in B16F10 melanoma cells. (Ec, Ef) Merge shows colocalization in yellow. (**F**) Quantification of colocalization was performed by Pearson's coefficient. Data are the mean ± SE (n = 12). *$p<0.05$ (Student's t-test). Scale bar represents 10 µm.

The online version of this article includes the following figure supplement(s) for figure 3:

**Figure supplement 1.** Kv1.3 CBD*less* does not localize to the plasma membrane in B16F10 melanoma cells.

respectively). Cells displaying severe effects concentrated Kv1.3 CBD*less* in multilamellar bodies, apoptotic phagocytic structures, and close to membranous whorls, which are characteristic of cell death (*Figure 6I–K*, respectively) and are consistent with the punctate pattern observed in confocal studies (*Figures 2* and *3*).

## Cav1-mitoKv1.3 functional link alleviates the pro-apoptotic activity of mitochondrial Kv1.3

As mentioned above, the physiological role of Kv1.3 in cell survival is complex because the channel exerts known, and apparently opposite, dual roles. However, the finding reported here using Kv1.3 CBD*less* may explain this duality. Functional Kv1.3 CBD*less*, which is unable to interact with Cav1, was trafficking to mitochondria rather than to the cell surface. Therefore, it is tempting to speculate that either the imbalance between plasma membrane Kv1.3 and mitoKv1.3 is the factor that contributes to apoptosis, or the lack of association with caveolin in mitochondria is what makes the difference. In

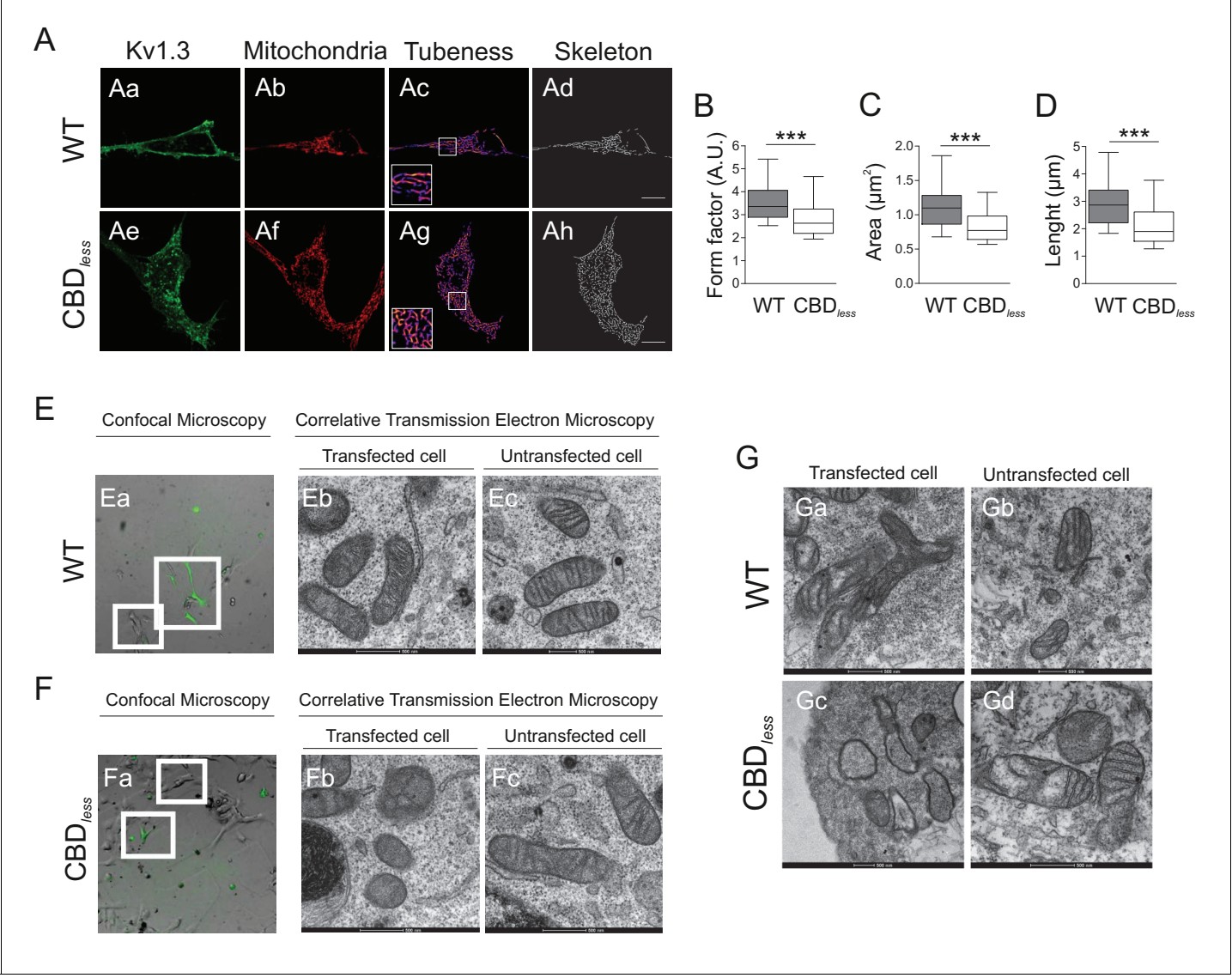

**Figure 4.** Kv1.3 CBD*less* targets mitochondria altering mitochondrial morphology. HEK 293 and B16F10 cells were transfected with Kv1.3YFP WT or Kv1.3YFP CBD*less*. (A) Representative confocal images of HEK-293 cells cotransfected with (Aa–Ad) Kv1.3YFP WT, (Ae–Ah) Kv1.3YFP CBD*less* (green) and pmitoRFP (red). Images were processed (tubeness (Ac, Ag) and skeleton (Ad, Ah)) to perform morphometric analysis (B–D) of mitochondria in Kv1.3 positive cells. Scale bar represents 10 µm. (B) The form factor (arbitrary units, A.U.) describes the particle shape complexity and is computed as the average (perimeter)$^2$/(4π·area). A circle corresponds to a minimum value of 1. (C) Average area of particles detected on the binary image. (D) The length of mitochondrial networks was measured as the average area of the skeletonized binary image. Data are the mean ± SE (n > 30). ***p<0.001 (Student's t-test). (E, F) Electron micrograph of B16F10 melanoma cells transfected with (Ea–Ec) Kv1.3YFP WT or (Fa–Fc) Kv1.3YFP CBD*less*. Cells were observed via confocal microscopy 3 days after transfection (Ea, Fa). Scale bar represents 40 µm. Next, cells were fixed and analyzed by correlative electron microscopy (Eb–Ec, Fb–Fc). (Eb, Fb) Transfected cells positive for Kv1.3 YFP (in green) from white squares in Ea and Fa. (Ec, Fc) Untransfected cells negative for Kv1.3 YFP from white squares in Ea and Fa. (G) Correlative electron micrograph of HEK 293 cells transfected with (Ga, Gb) Kv1.3YFP WT or (Gc, Gd) Kv1.3YFP CBD*less*. (Ga, Gc) Transfected cells positive for Kv1.3 YFP. (Gb, Gd) Untransfected cells negative for Kv1.3 YFP. Note the lack of mitochondrial cristae and the presence of swollen mitochondria in Kv1.3YFP CBD*less* transfected cells (Gc). Images are representative of three independent experiments. Scale bar represents 500 nm.

any case, interaction with caveolin would be at the onset of both scenarios. Therefore, because caveolin exerts anti-apoptotic effects (*Shiroto et al., 2014*; *Schilling et al., 2018*), we wondered whether Cav1, by association with the channel, may modulate the role of Kv1.3 in mitochondria. Similar to Kv1.3, Cav1 is also present in mitochondria (*Figure 7A*), and both proteins are located in close proximity (*Figure 7B,C*). To evaluate the effects of the Cav1 interaction on mitoKv1.3, we used

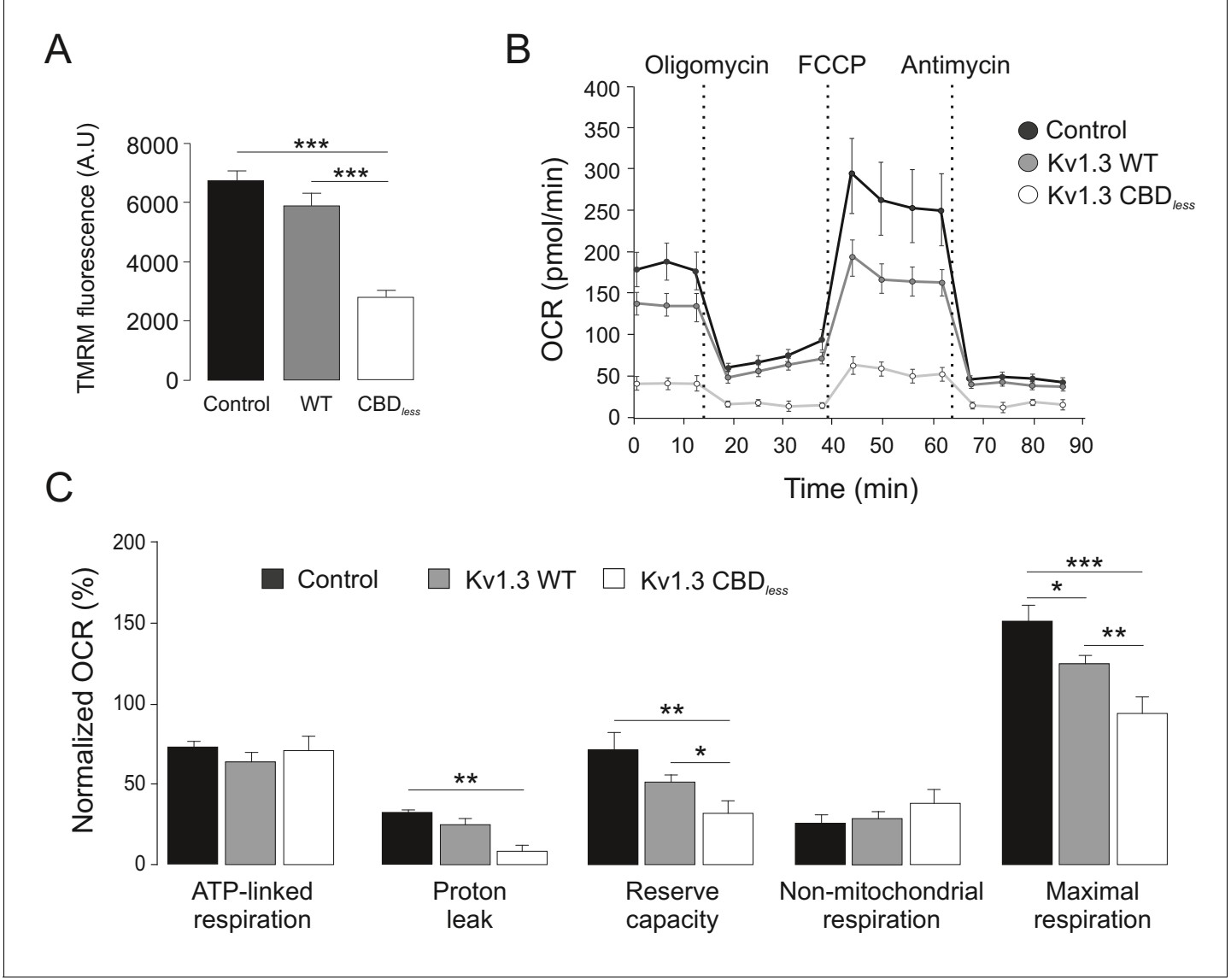

**Figure 5.** Kv1.3 CBD$_{less}$ severely impairs mitochondrial function. HEK-293 cells were transfected with Kv1.3YFP WT and Kv1.3YFP CBD$_{less}$. Non-transfected cells were used as a control. (**A**) Mitochondrial membrane potential was determined by tetramethyl rhodamine methyl ester (TMRM) fluorescence. Positive transfected cells (separated by sorting) were incubated with TMRM and analyzed by confocal microscopy. Data are the mean ± SE (n = 3). ***p<0.001 (one-way ANOVA). A.U, arbitrary units. (**B**) The oxygen consumption rate (OCR) of HEK293 cells transfected with Kv1.3YFP WT or Kv1.3YFP CBD$_{less}$ in the presence of 2 μg/ml oligomycin (ATPase synthase inhibitor), 200 nM FCCP (respiratory chain uncoupler), 1 μM antimycin (complex III blocker). (**C**) Normalized OCR parameters (%) extracted from (**B**). Data are the mean ± SE (n = 3). *p<0.05; **p<0.01; ***p<0.001 (one-way ANOVA). Black bar/circle, non-transfected control cells; Gray bar/circle, Kv1.3 WT; white bar/circle, Kv1.3 CBD$_{less}$.

several different and complementary cell models. Jurkat T lymphocytes are normally deficient in caveolin (Jurkat Cav−), but express endogenous Kv1.3 (*Szabò et al., 2005*; *Figure 7D*). However, we selected a clone of Jurkat cells, which did express Cav1 (Jurkat Cav+). In this context, evidence indicates that some leukocytes, that do not initially express caveolin, may express the protein under certain states of activation (*Hatanaka et al., 1998*). In Jurkat Cav−, no differences were observed in apoptosis when Kv1.3 WT or Kv1.3 CBD$_{less}$ were overexpressed (*Figure 7D,E*). However, Kv1.3 WT cells exhibited less apoptosis than cells expressing Kv1.3 CBD$_{less}$ in Jurkat Cav+ (*Figure 7E*). Next, we used 3T3-L1 preadipocytes, which endogenously express both Cav1 and Kv1.3 (*Pérez-Verdaguer et al., 2018*). By knocking-down Cav1 expression (Cav−), 3T3-L1 cells became more sensitive to apoptosis (*Figure 7F,G*), recapitulating the Kv1.3 CBD$_{less}$ effects observed in HEK 293 and

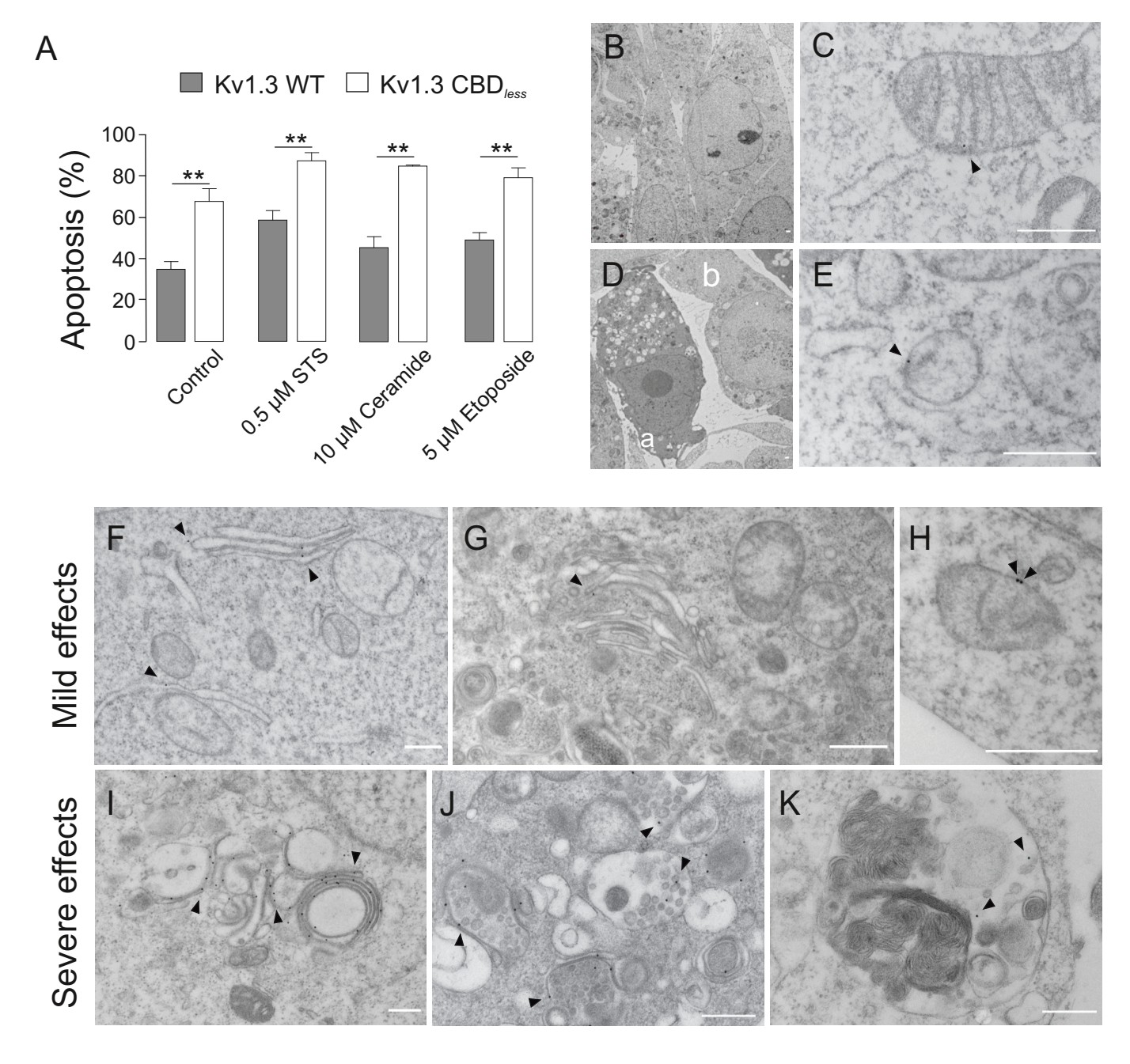

**Figure 6.** Kv1.3 CBD$_{less}$ sensitizes cells to apoptosis. HEK-293 cells were transfected with Kv1.3YFP WT or Kv1.3YFP CBD$_{less}$. (**A**) Flow cytometry analysis evaluating apoptosis by Annexin V staining. Cells were cultured for 24 hr in the absence (control) or the presence of different pro-apoptotic compounds (0.5 µM staurosporine [STS], 10 µM ceramide, and 5 µM etoposide). Transfected cells were sorted and the % of Annexin V-positive cells was calculated. Gray bars, Kv1.3 WT. White bars, Kv1.3 CBD$_{less}$. Data are the mean ± SE (n = 3). **p<0.01 (Student's t-test). (**B–K**) Electron micrographs showing ultrastructural features of HEK-293 cells transfected with Kv1.3YFP WT (**B, C**) or Kv1.3YFP CBD$_{less}$ (**D–K**). Kv1.3 was immunolabeled with 18 nm diameter gold particles. Arrowheads show Kv1.3. (**B**) Normal appearance of organelles in a Kv1.3YFP WT-transfected cell. (**C**) Expression of Kv1.3YFP WT at the inner mitochondrial membrane. (**D**) Kv1.3YFP CBD$_{less}$ triggered either severe (a) or mild (b) apoptotic cell phenotypes in cells. (**E**) Kv1.3 CBD$_{less}$ at the mitochondrial membrane. Note the absence of mitochondrial cristae. (**F–H**) Localization of Kv1.3 CBD$_{less}$ in cells affected with a mild apoptotic phenotype. (**F**) Kv1.3 CBD$_{less}$ at the ER. (**G**) Kv1.3 CBD$_{less}$ at the Golgi apparatus. (**H**) Kv1.3 CBD$_{less}$ at the mitochondrial membrane. (**I–K**) Localization of Kv1.3 CBD$_{less}$ in cells affected with a severe apoptotic phenotype. (**I**) Notable accumulation of Kv1.3 CBD$_{less}$ in membranes surrounding lysosomes or autolysosomes. (**J**) Intense staining of Kv1.3 CBD$_{less}$ at multivesicular bodies. (**K**) Kv1.3 CBD$_{less}$ in vacuole containing membrane whorls. Bars represent 500 nm.

*Figure 6 continued on next page*

*Figure 6 continued*

The online version of this article includes the following figure supplement(s) for figure 6:

**Figure supplement 1.** Flow cytometry evaluates apoptosis by Annexin V staining.

**Figure supplement 2.** The accumulation of Kv1.3 in mitochondria is not responsible for the increase in apoptosis.

**Figure supplement 3.** Kv1.3 CBD$_{less}$ triggered no ER-stress.

B16F10 melanoma cells. Our data indicated that altering the functional crosstalk between Cav1 and Kv1.3 plays a crucial role in determining the sensitivity of cells to apoptosis.

Finally, we isolated CD4+ T lymphocytes from blood of human donors. As expected, human T lymphocytes, which express membrane lipid raft integral proteins such as flotillin, express endogenously Kv1.3, but not Cav 1 (*Figure 8A*). Confocal experiments, performed in Kv1.3 WT and CBD$_{less}$ transfected cells, indicated that both Kv1.3 channels shared similar plasma membrane (Mb) and mitochondrial (mito) colocalization (*Figure 8B–E*). Morphometric analysis of Kv1.3 WT and CBD$_{less}$ T cells indicated that mitochondria were similarly affected by the expression of either channel (*Figure 9*). Thus, both channels triggered an increase in mitochondrial length and form factor, whereas mitochondrial area diminished in CD4+ lymphocytes. The expression of Kv1.3 WT and CBD$_{less}$ decreased also the mitochondrial membrane potential (*Figure 10A*), which is concomitant to similar levels of apoptosis (*Figure 10B*). Interestingly, the introduction of external Cav 1 in primary CD4 + cells (Cav+) partially counteracted apoptosis solely in cells expressing Kv1.3 CBD$_{less}$. Our data from primary CD4+ human T lymphocytes showed that the presence of Cav 1, which interacts with Kv1.3 WT but not CBD$_{less}$, partially protected cells against apoptosis and further confirmed what obtained with Jurkat T cells and 3T3-L1 cells.

## Discussion

Kv1.3 plays important but apparently contrasting roles in the cell physiology because Kv1.3 in the plasma membrane supports proliferation, while mitoKv1.3 sensitizes cells to apoptosis. Furthermore, caveolin is also involved in both pro- and anti-apoptotic events participating in the regulation of cell survival and in cancer protection. Our results indicate that the interaction of Kv1.3 with Cav1 has important physiological consequences for controlling apoptosis. Cav1 association, via the N-terminal located CBD of Kv1.3, drives the channel to the plasma membrane. Altering the CBD impairs membrane targeting, promoting the Kv1.3 intracellular retention and mitochondrial accumulation. In this way, we were able to distinguish the effects of intracellular/mitochondrial Kv1.3 from those of the plasma membrane channel. Once inside mitochondria, the CBD$_{less}$ channel facilitates apoptosis. In the presence of Kv1.3, either the depletion of caveolin or the lack of Cav1 binding favors apoptosis, suggesting that the Cav1 functional interaction with the channel facilitates apoptotic resistance. Therefore, we identified that Cav1 associates with Kv1.3, thereby modulating the pro-apoptotic effects of mitochondrial Kv1.3 channels. This observation is key for the understanding how these two proteins can reciprocally regulate their role in cancer progression, with significant implications for anti-cancer therapy.

Acute inhibition of mitoKv1.3 transient hyperpolarizes IMM inducing ROS release. Subsequent PTP opening leads to loss of mitochondrial integrity, IMM depolarization, swelling, and cytochrome c release. Overexpression of mitochondrial K$^+$ transporting pathways drives the influx of depolarizing K$^+$ into the matrix triggering mitochondrial depolarization as well as changes in ultrastructure (*Paggio et al., 2019*). Thus, acute changes in the mitochondrial membrane potential, upon block of the channel, induce ROS and PTP opening, while overexpression of mitochondria-located Kv1.3, that is more prominent in the case of CBDless mutant or in the absence of Cav (for both WT and CBD$_{less}$ Kv1.3), causes sustained depolarization that sensitizes cells to apoptotic stimuli. PTP opening further depolarizes IMM and reduces respiration due to swelling and loss of cytochrome c. Here we show that overexpression of WT or CBD$_{less}$ Kv1.3 equally depolarized mitochondria and caused apoptosis in primary T cells, which lack endogenous Cav. Interestingly, while overexpression of Kv1.3 promotes apoptosis, the channel deficiency renders the cells resistant to apoptosis (*Szabó et al., 2008*). Thus, similarly to the ATP-dependent K$^+$ channel of pancreatic β cells (*Miki et al., 1998*), pharmacological

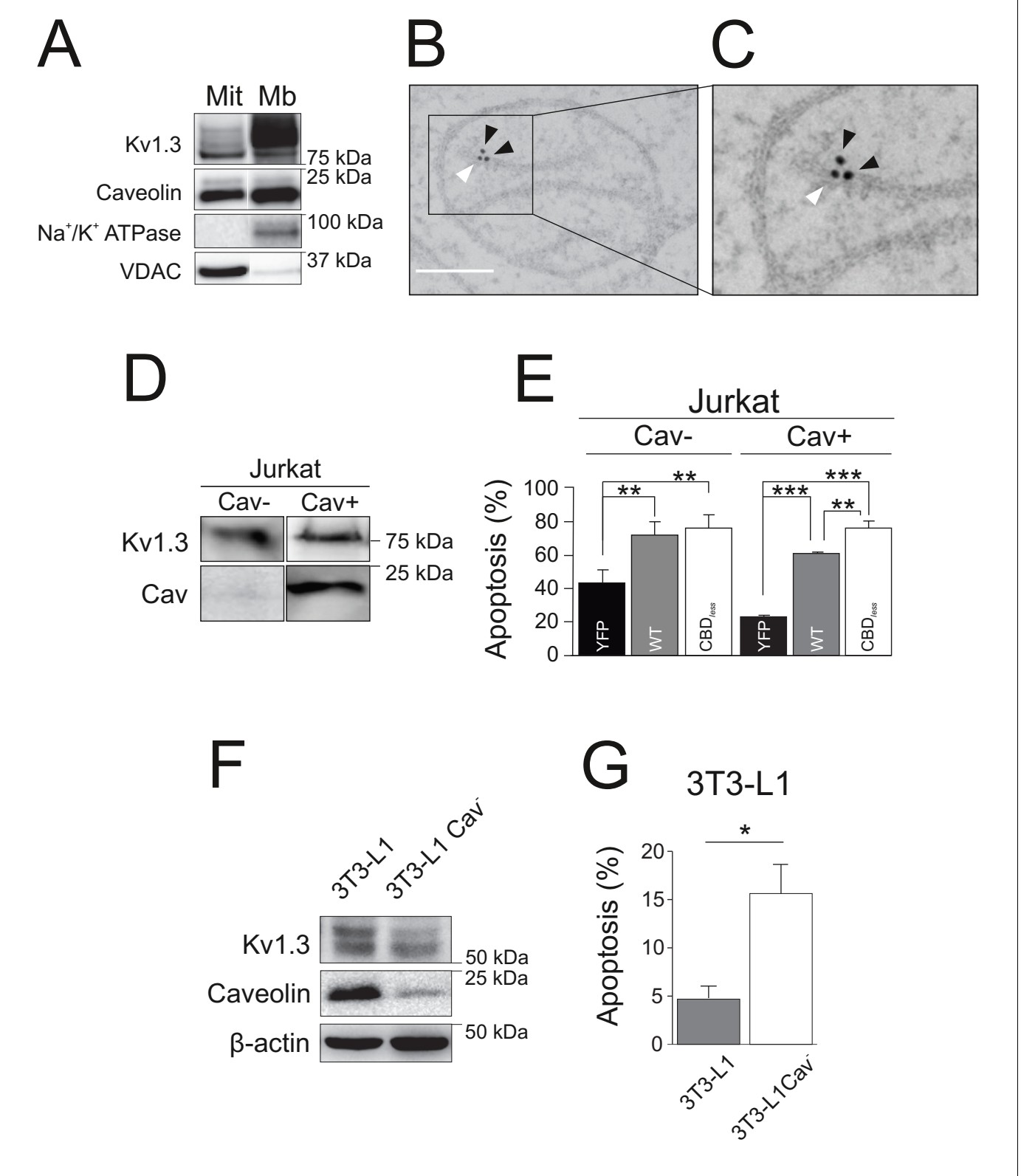

**Figure 7.** Caveolin modulates the pro-apoptotic activity of Kv1.3. (**A**) Subcellular fractionation was used to isolate mitochondrial (Mit) and plasma membrane (Mb) fractions in HEK-293 cells transfected with Kv1.3YFP WT. Samples were immunoblotted for GFP (Kv1.3), Caveolin, $Na^+/K^+$ ATPase, and VDAC. (**B, C**) Electron micrographs showing HEK-293 cells transfected with Kv1.3YFP WT. Kv1.3 was immunolabeled with 18 nm gold particles (black arrowheads) and Cav1 with 12 nm gold particles (white arrowhead). The square inset in (**B**) indicates the zoomed in region in (**C**). Scale bars represent

*Figure 7 continued on next page*

*Figure 7 continued*

500 nm. (D) Regular human Jurkat T lymphocytes express Kv1.3 and a negligible amount of endogenous Cav1 (Cav$^-$). In addition, a Jurkat cell line with notable expression of Cav1 was selected (Cav+). (E) Jurkat cells (Cav$^-$ and Cav$^+$) were electroporated with Kv1.3YFP WT or Kv1.3YFP CBD$_{less}$. After 24 hr, apoptosis was assessed by Annexin V staining with flow cytometry. Black bar, cells electroporated with YFP; gray bar, Kv1.3 YFP WT; white bar, Kv1.3YFP CBD$_{less}$. (F) Mouse 3T3-L1 and 3T3-L1 Cav$^-$ preadipocytes were analyzed for the expression of endogenous Cav1 and Kv1.3. β-actin was used as a loading control. (G) Flow cytometric analysis quantifying apoptosis by Annexin V on 3T3-L1 (gray bar) and 3T3-L1 Cav$^-$ (white bar) preadipocytes. Note that the amount of Cav1 exerted notable effects on the Kv1.3-related apoptosis in native 3T3-L1 cells. Values are the mean ± SE of 3–6 independent experiments. *p<0.05; **p<0.01; ***p<0.01 (one-way ANOVA).

block of Kv1.3 does not yield the same physiological effects of channel downregulation, as the block triggers a series of signaling events that cannot be induced in cells lacking Kv1.3.

Cav interacts with the CBD of target signaling proteins through its CSD. Mutating the CSD of Cav1 modulates migration of cancerous cells by affecting interaction with signaling partners (*Okada et al., 2019*). Kv1.3, by interaction with Cav1, targets to lipid raft microdomains. These observations are consistent with our previous data, showing that the presence of Kv1.3 in lipid rafts was dependent on Cav1 expression (*Pérez-Verdaguer et al., 2016a*). Evidence supports that CBDs should contain solvent exposed aromatic residues (*Byrne et al., 2012*). Our data support this claim because, in our molecular simulation, the Kv1.3 CBD is exposed and located close to transmembrane domains, which favor physical interactions with Cav1. Aromatic residues are involved in protein folding. Therefore, because of the proximity of the CBD to the channel tetramerization domain (T1), we assessed Kv1.3 CBD$_{less}$ functionality. Kv1.3 CBD$_{less}$ forms functional tetramers but with altered electrophysiological properties. The main feature is the reduction of current intensity, probably due to impaired surface expression. In fact, we have previously shown that the absence of Cav1 expression reduces the half-life, cell surface expression, and current amplitude of Kv1.3 (*Pérez-Verdaguer et al., 2016a*). Kv1.3 CBD$_{less}$ showed a hyperpolarizing shift in the steady-state activation and altered inactivation kinetics. These observations are not surprising because the lipid composition of the plasma membrane modulates the activity, kinetics, and voltage-dependence of ion channels (*Levitan et al., 2010*; *Brini et al., 2018*; *Poveda et al., 2017*; *Zakany et al., 2019*). Furthermore, the biophysical properties of Kv1.3 CBD$_{less}$ might be due to multiple additional factors, including changes in membrane lipid composition that occur during apoptosis (*Tepper et al., 2000*). Altogether, our data would support that Kv1.3 surface expression and thus the current density are notably dependent on its Cav1 interaction.

As mentioned above, the lack of interaction between Kv1.3 CBD$_{less}$ and Cav1 drastically impaired the cell surface targeting of the channel. However, Kv1.3 CBD$_{less}$ kept targeting to mitochondria, and this occurred before Golgi processing, given that the mitochondria-targeted channel is not glycosylated. This fact raises up an interesting scenario because we separated the effects of mitochondrial Kv1.3 from those of the plasma membrane channel. Expression of Kv1.3 CBD$_{less}$ notably sensitized the cells to apoptosis, most likely due to the preferential localization of this mutated channel to the mitochondria. Indeed, by itself, mitochondrial Kv1.3 CBD$_{less}$ dramatically altered mitochondrial morphology (i.e., lower cristae density) and function, triggering apoptosis. In addition, expression of Kv1.3 CBD$_{less}$ caused mitochondrial network fragmentation, something typically observed in apoptotic cells, which show a high fission-to-fusion ratio (*Xie et al., 2018*). In general, mitochondrial fission regulation couples mitochondrial dynamics/morphology to the cellular energetic state (*Giacomello et al., 2020*). Fission, induced by AMPK-mediated phosphorylation (AMP), senses adapting mitochondrial function and dynamics. AMPK facilitates mitochondrial fission downstream of mitochondrial dysfunction caused, for example, by an impaired mitochondrial respiration. This mechanism facilitates apoptosis in cells with severely dysfunctional mitochondria (*Toyama et al., 2016*). Apoptosis can be triggered by the permeabilization of the outer mitochondrial membrane and the prolonged depolarization of the IMM (dissipation of $\Delta\psi_m$ in IMM), which causes the release of pro-apoptotic factors from the mitochondrial intermembrane space to the cytosol (*Zorova et al., 2018*). We can therefore explain the ability of Kv1.3 CBD$_{less}$ triggered apoptosis by the observation that mitochondria were significantly depolarized and mitochondrial respiration was highly impaired, suggesting that cells were undergoing apoptosis. We propose that the absence of Cav1 interaction with mitoKv1.3 favors the pro-apoptotic effects of Kv1.3. The results obtained in human CD4+ lymphocytes, Jurkat T cells, and 3T3-L1 preadipocytes support this

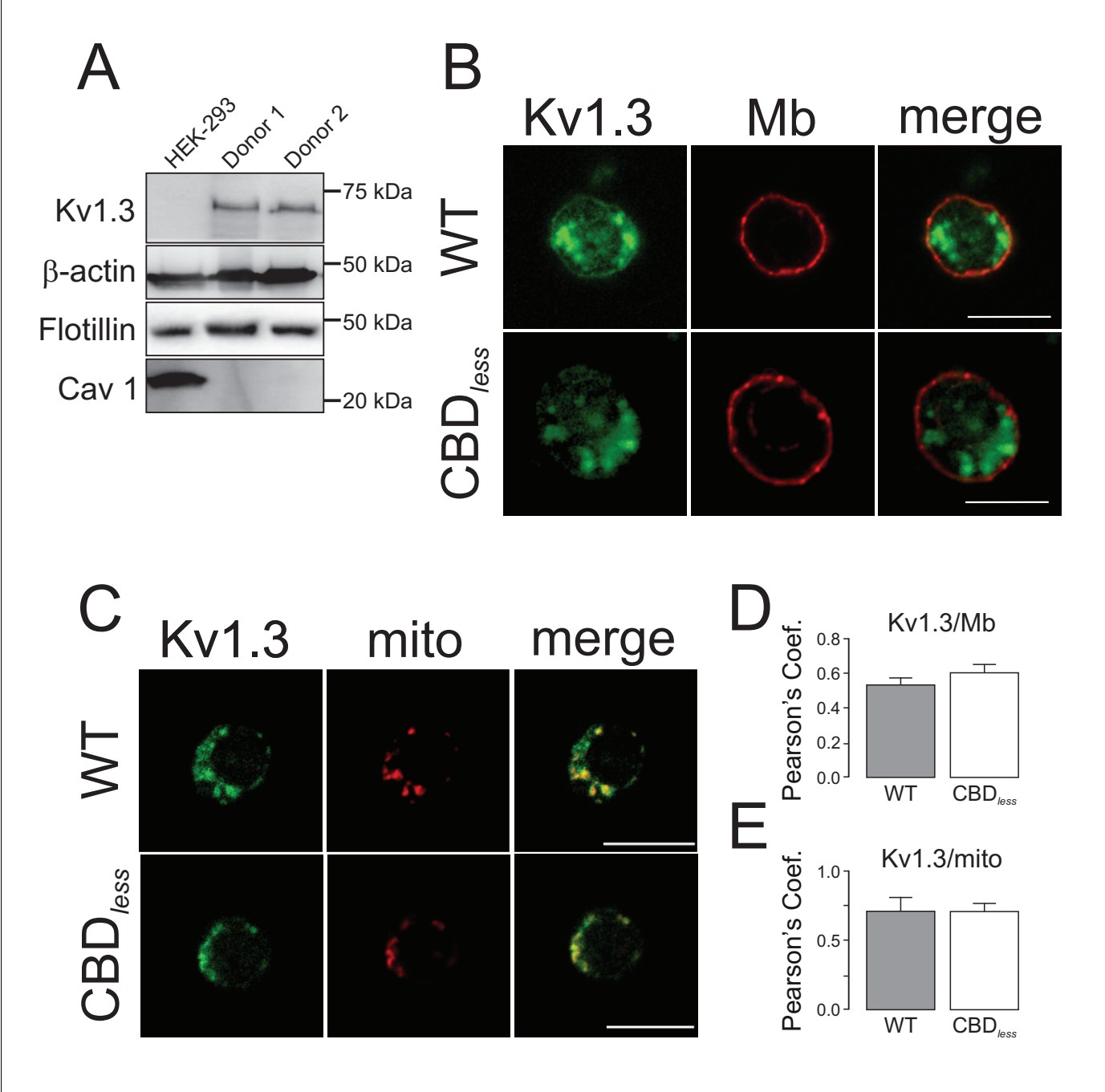

**Figure 8.** Kv1.3 colocalizes with plasma membrane and mitochondria in primary human T lymphocytes. CD4+ lymphocytes were isolated from human blood as indicated in Materials and methods. (**A**) Representative western blot from HEK 293 cells and T lymphocytes samples from two independent human donors showing differential protein expression of Kv1.3, Flotillin, and Cav1. β-Actin was a loading control. (**B**) Representative confocal images of Kv1.3 colocalization in plasma membrane (Mb) from Kv1.3YFP WT and Kv1.3YFP CBD$_{less}$-transfected cells. WGA stained plasma membrane. (**C**) Representative confocal images of Kv1.3 colocalization in mitochondria (mito) from Kv1.3YFP WT and Kv1.3YFP CBD$_{less}$ expressing cells. MitoTracker was used for mitochondrial staining. Scale bar represents 10 μm. Quantification of Kv1.3/Mb (**D**) and Kv1.3/mito (**E**) colocalization was performed by Pearson's coefficient. Data are the mean ± SE (n > 20), Student's t-test. Gray bars, Kv1.3YFP WT cells; white bars, Kv1.3YFP CBD$_{less}$ cells.

hypothesis. In T-lymphocytes, which lack of endogenous Cav1 expression, no differences in apoptosis between cells expressing Kv1.3 WT and Kv1.3 CBD$_{less}$ were found. Similarly, silencing of Cav1 expression in 3T3-L1 preadipocytes increased the apoptosis. However, the presence of Cav1 exhibited anti-apoptotic properties but solely in cells expressing the wild-type channel.

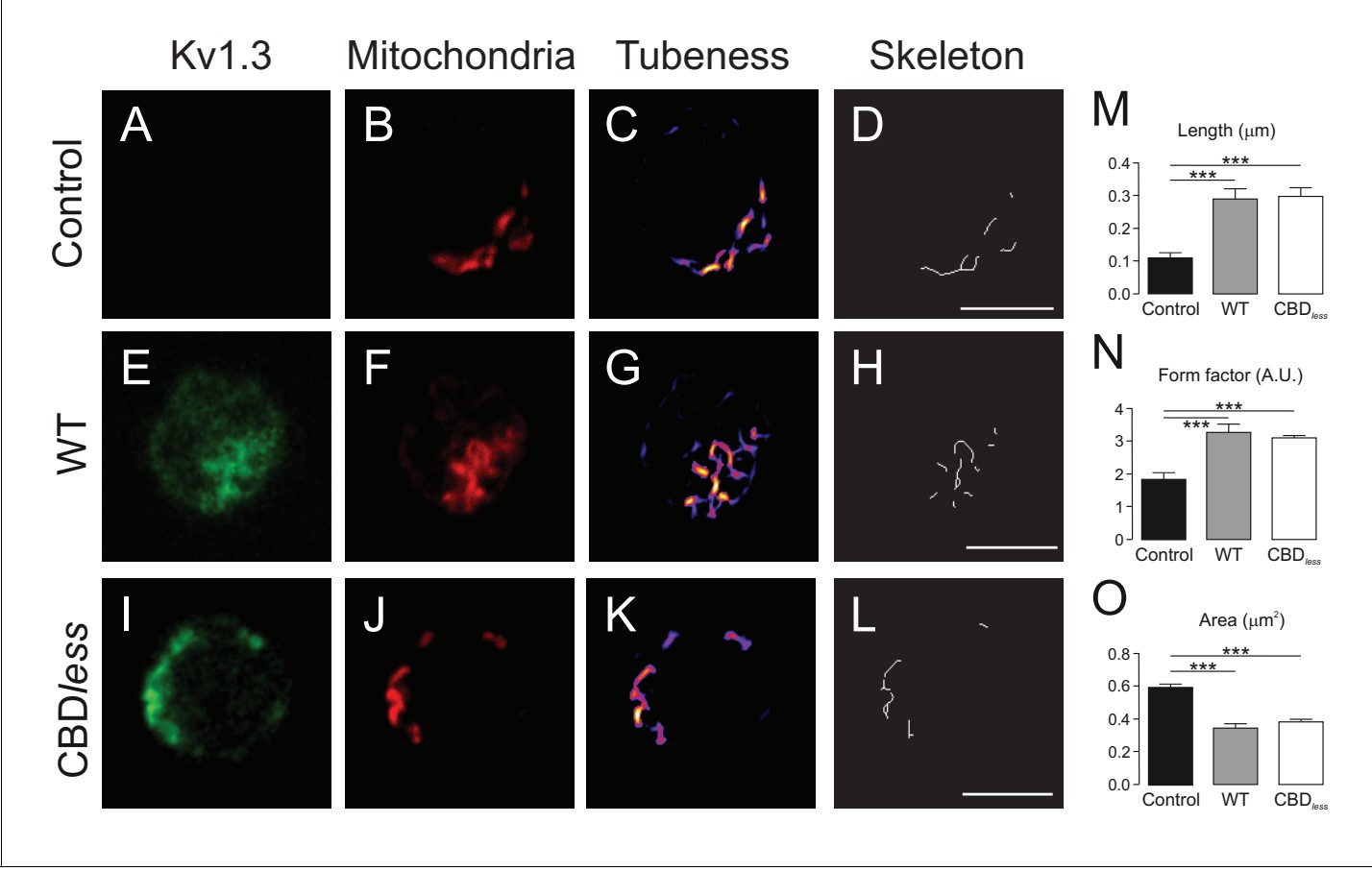

**Figure 9.** The expression of Kv1.3YFP WT and Kv1.3YFP CBD*less* in T lymphocytes alters the mitochondrial morphology. Human CD4+ T lymphocytes were transfected with Kv1.3YFP WT or Kv1.3YFP CBD*less*. (**A–D**) Representative confocal images of non- transfected T cells. (**E–H**) T lymphocytes transfected with Kv1.3YFP WT. (**I–L**) T lymphocytes transfected with Kv1.3YFP CBD*less*. (**A, E, and I**) Kv1.3YFP (green); (**B, F, and J**) MitoTracker (red). Images were processed (tubeness (**C, G, and K**) and skeleton (**D, H, and L**)) to perform morphometric analysis (**M, N, and O**) of mitochondria. Scale bar represents 10 µm. (**M**) The length of mitochondrial networks was measured as the average area of the skeletonized binary image. (**N**) The form factor (arbitrary units, A.U.) describes the particle shape complexity and is computed as the average (perimeter)$^2$/(4$\pi$·area). A circle corresponds to a minimum value of 1. (**O**) Average area of particles detected on the binary image. Data are the mean ± SE (n > 20). ***p<0.001 (Student's t-test).

In summary, our data indicate that the physiological role of Kv1.3 is highly dependent on its interaction with Cav1. The Kv1.3 interaction with Cav1 can drive the channel either to the membrane and support proliferation or to mitochondria. MitoKv1.3 sensitizes cells to apoptosis, in agreement with our previous observation (*Szabó et al., 2008*), and possibly, mitochondrial Kv1.3 interaction with Cav1 modulates the pro-apoptotic effects of the channel. Therefore, the balance exerted by these two complementary mechanisms would fine-tune the physiological role of Kv1.3 during cell survival or apoptosis. Although a direct interaction of these proteins in mitochondria has not been confirmed, and warrants further investigation, evidence suggests that apoptosis is dependent on mitoKv1.3–caveolin functional axis. Our work has important implications not only in the understanding of Kv1.3-dependent cancer progression but possibly also in metabolic diseases, where Cav1 and Kv1.3 both play an important role. Our data would suggest an essential role for the caveolin–Kv1.3 axis during tumorigenesis and apoptosis.

## Materials and methods

### Expression plasmids and site-directed mutagenesis

T.C. Holmes (University of California, Irvine, CA) provided the rat Kv1.3 in a pRcCMV construct. The channel was subcloned into pEYFP-C1 and pECerulean-C1 (Clontech). All Kv1.3 mutants were

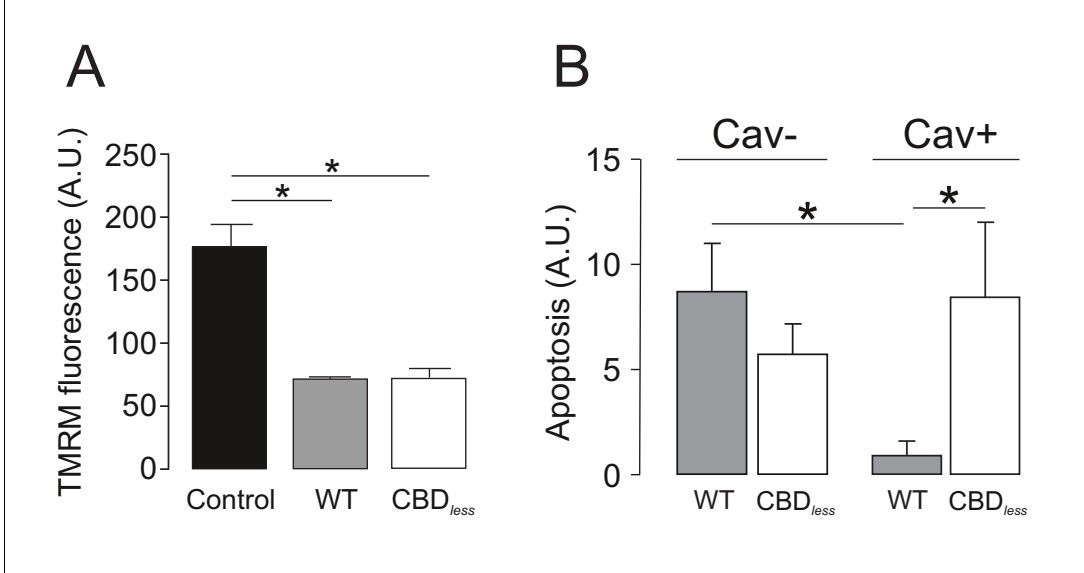

**Figure 10.** Caveolin-1 protects from apoptosis when associated with Kv1.3 in primary human T lymphocytes. Human CD4+ T lymphocytes were transfected with Kv1.3YFP WT or Kv1.3YFP CBD$_{less}$ and the mitochondrial membrane potential (TMRM) and apoptosis were measured. YFP-transfected cells were used as a control. (**A**) Mitochondrial membrane potential was determined by tetramethyl rhodamine methyl ester (TMRM) fluorescence. Cells were incubated with TMRM and analyzed by flow cytometry. A.U, arbitrary units. (**B**) T cells were electroporated with Kv1.3YFP WT or Kv1.3YFP CBD$_{less}$ with (Cav+) or without (Cav−) Cav1 Cerulean. After 24 hr, transfected cells were sorted and apoptosis was assessed by Annexin V staining with flow cytometry. The level of apoptosis in arbitrary units (A.U.) was measured in each group by resting the value of basal apoptosis in cells transfected with YFP in the presence (Cav−) or the absence (Cav−) of Cav1. Black bar, cells electroporated with YFP; gray bar, Kv1.3 YFP WT; white bar, Kv1.3YFP CBD$_{less}$. Cav−, regular CD4+ cells without Cav 1; Cav+, T cells transfected with Cav1 Cer. Data are the mean ± SE (n = 5–7). *p<0.05 (one-way ANOVA).

generated in the pEYFP-Kv1.3 channel using a QuikChange site-directed mutagenesis kit (Agilent Technologies). pEYFP-Kv1.3CBD$_{less}$ was subcloned into pECerulean-C1. For oocyte injection, Kv1.3 and Kv1.3CBD$_{less}$ were subcloned into pcDNA3 and were placed under the control of a T7 promoter and then the cRNA was synthetized. J.R. Martens (University of Florida Medical School) provided the rat Caveolin 1 (Cav1) in pECerulean-C1. Cav1 was cloned into pcDNA3. The plasma membrane marker Akt-PH-pDsRed (pDsRed-tagged pleckstrin homology (PH) domain of Akt) was a kind gift of F. Viana (Universidad Miguel Hernández, Spain). The ER marker (pDsRed-ER) was obtained from Clontech. The mitochondrial marker (pmitoRFP) was constructed by fusing the mitochondrial transit sequence of the human isovaleryl coenzyme A dehydrogenase to the N-terminus of RFP (pDsRed1-N1, Clontech). Constructs were verified by sequencing.

## Cell culture and drugs

HEK 293 cells (ATCC) were grown in Dulbecco's modified Eagle medium (DMEM) containing 10% fetal bovine serum (FBS) and 100 U/ml penicillin/streptomycin (Gibco). In some experiments, a HEK 293 Cav$^−$ (lentiviral depletion of Cav1) was used (*Pérez-Verdaguer et al., 2016a*). B16F10 cells (ATCC) were grown in minimum essential media (Thermo Fisher Scientific) supplemented with 10 mM HEPES buffer (pH 7.4), 10% FBS, 100 U/ml penicillin G, 0.1 mg/ml streptomycin, and 1% nonessential amino acids (100× solution; Thermo Fisher Scientific). Transient transfection was performed following the manufacturer's instructions using Lipotransfectin (Attendbio), for HEK 293 cells, or TransIT-LT1 Transfection Reagent (Mirus), for B16F10 cells. Transfections were performed when cells were nearly 80% confluent. Jurkat human T lymphocytes (ATCC) were cultured in RPMI media containing 10% FBS and transfected using the Gene Pulser II Electroporation system and YFP-expressing cells were sorted using a FACSAria FUSION (BD Bioscience) instrument. 3T3-L1 preadipocytes (ATCC) were cultured in DMEM containing 10% NCS in a 7% $CO_2$ atmosphere. All drugs were dissolved in dimethyl sulfoxide (DMSO) and diluted in DMEM. The final concentration of DMSO was <0.5% in all assays. All cell lines were routinely tested to be mycoplasma free.

## Isolation of T-cell subsets, cell culture, and T-cell blast generation

Human CD4+ T-cell subsets were isolated from peripheral whole blood using negative selection Rosette Sep kit from STEMCELL Technologies. Human T lymphocytes were cultured at 37°C, 5% $CO_2$, in RPMI 1640 medium (Life Technologies) supplemented with 10% FCS, 1% glutamine, 1% penicillin–streptomycin (Gibco), 1× Non-Essential Amino Acids Solution (Thermo Fisher Scientific), 10 mM HEPES (Life Technologies), and 50 U/ml IL-2 (Bionova). To generate T-cell blasts, the Dynabeads Human T-Activator CD3/CD28 for T-cell expansion and activation kit (Life Technologies) was used following manufacturer's instructions. Human T-cell blasts were used after 6–7 days of expansion protocol. No IL-2 was supplemented in media the day before an experiment. In some experiments, pEYFP-Kv1.3 WT, pEYFP-Kv1.3CBD$_{less}$, and pECerulean-Cav1 were electroporated into human CD4 + lymphocytes as abovementioned.

The protocol was reviewed and approved by the Ethics Committee of the Universitat de Barcelona and the Banc de Sang i Teixits de Catalunya (BST). Institutional Review Board (IRB00003099). All procedures followed the rules of the Declaration of Helsinki Guidelines. All donors signed a written informed consent, and samples were totally anonymous and untraceable.

## Raft isolation

Low-density, Triton-insoluble complexes were isolated as previously described (*Pérez-Verdaguer et al., 2016a*). Briefly, after three washes in phosphate-buffered saline (PBS), cells were homogenized in 1 ml of 1% Triton X-100 MBS (150 mM NaCl, 25 mM 2-morpholinoethanesulfonic acid 1-hydrate, pH 6.5) supplemented with 1 µg/ml aprotinin, 1 µg/ml leupeptin, 1 µg/ml pepstatin, and 1 mM phenylmethylsulfonyl fluoride to inhibit proteases. Sucrose in MBS was added to a final concentration of 40%. A 5–30% linear sucrose gradient was layered on top and further centrifuged (39,000 rpm) for 20–22 hr at 4°C in a Beckman SW41Ti swinging rotor. Gradient fractions (1 ml) were collected from the top and analyzed by western blot.

## Purification of mitochondria

Mitochondria from HEK 293 cells were purified by differential centrifugation (adapted from *Wieckowski et al., 2009*). Briefly, 80% confluent cells were trypsinized and washed twice with PBS without $Ca^{2+}$ and centrifuged at 600 × g for 10 min. Cells were homogenized in initial buffer 1 (225 mM mannitol, 75 mM sucrose, 0.1 mM EGTA, 30 mM Tris, pH 7.4) and centrifuged again at 600 × g for 10 min to remove unlysed cells and nuclei. The supernatant was centrifuged at 7000 × g 10 min. The mitochondria-containing pellet was suspended in initial buffer 2 (225 mM mannitol, 75 mM sucrose, 30 mM Tris, pH 7.4) and centrifuged again at 7000 × g. The suspension of the pellet was repeated and centrifuged at 10,000 × g to obtain a purified mitochondrial fraction. The supernatant from the first 7000 × g centrifugation was then centrifuged at 20,000 × g for 30 min and the pellet contained the membranous fraction. Mitochondrial and membranous fractions were suspended in 50 µl of initial buffer 2. All centrifugations were performed at 4 °C.

## Protein extraction, coimmunoprecipitation, biotinylation of cell surface proteins, and western blot analysis

Cells were washed in cold PBS, lysed on ice with NHG solution (1% Triton X-100, 10% glycerol, 50 mM HEPES pH 7.2, 150 mM NaCl) supplemented with 1 µg/ml aprotinin, 1 µg/ml leupeptin, 1 µg/ml pepstatin, and 1 mM phenylmethylsulfonyl fluoride to inhibit proteases. Homogenates were centrifuged at 16,000 × g for 15 min, and the protein content was measured using the Bio-Rad Protein Assay. For immunoprecipitation, samples were precleared with 30 µl of protein A-sepharose beads for 2 hr at 4°C with gentle mixing and the beads were then removed by centrifugation at 1000 × g for 30 s at 4°C as part of the coimmunoprecipitation procedures. Meanwhile, 50 µl of protein A-sepharose beads were incubated in 500 µl of NGH in the presence or in the absence of an anti-caveolin antibody (4 ng/µg protein) at 4°C with gentle agitation and washed three times to obtain antibody-bound A-sepharose beads. The precleared samples were then incubated overnight at 4°C with antibody-bound A-sepharose beads. Finally, supernatants were removed by centrifugation at 1000 × g for 30 s at 4°C, and beads were washed four times with NHG and resuspended in 100 µl of Laemmli SDS buffer.

Three oocytes were placed into a 1.5 ml Eppendorf tube and homogenized by pipetting in 100 µl of homogenization buffer (20 mM Tris–HCl pH 7.6, 0.1 M NaCl, 1% Triton X-100) with protease inhibitor cocktail. Homogenates were incubated for 20 min at 4°C to solubilize membrane proteins and centrifuged at 10,000 × g for 2 min at 4°C. Supernatants were transferred to a new tube, and protein content was determined using the Bio-Rad Protein Assay (Bio-Rad).

Cell surface biotinylation was determined with the Pierce Cell Surface Protein Isolation Kit (Pierce) following manufacturer's instructions. Briefly, cell surface proteins were labeled with sulfosuccini-midyl-2-(biotinamido)ethyl-1,3-dithiopropionate (Sulfo-NHS-SS-biotin; Pierce). Then, cells were treated with lysis buffer, and clear supernatant was reacted with immobilized NeutrAvidin gel slurry in columns (Pierce) to isolate surface proteins. Protein samples (50 µg), raft fractions (50 µl), mitochondria and membranous fractions (50 µl), and immunoprecipitates were boiled in Laemmli SDS loading buffer and separated by 10% SDS–PAGE. Next, samples were transferred to PVDF membranes (Immobilon-P, Millipore) and blocked with 5% dry milk-supplemented with 0.05% Tween 20 in PBS. The filters were then immunoblotted with specific antibodies: anti-GFP (1:500, Roche), anti-caveolin (1:250, BD Biosciences), anti-Kv1.3 (1:200, Neuromab), anti-clathrin (1:1000, BD Biosciences), anti-flotillin (1:500, BD Biosciences), anti-β actin (1:50,000, Sigma), anti-Na$^+$/K$^+$ ATPase (Developmental Studies Hybridoma Bank, The University of Iowa), anti-VDAC (1:5000, Calbiochem), anti-GRP78 (1:1000, Cell Signaling Technology), anti-XBP1 (1:1000, Abcam), anti-ATF4 (1:500, Santa Cruz Biotechnologies), anti-eIF2α (1:1000, Abcam), and anti-eIF2α pS51 (1:1000, Abcam). Finally, membranes were washed with 0.05% Tween 20 in PBS and incubated with horseradish peroxidase conjugated secondary antibodies (Bio-Rad).

## Immunocytochemistry and confocal imaging

Cells seeded on poly-D-lysine-treated coverslips were used 24 hr later for transfection. Cells were washed with PBS and fixed (only HEK 293 cells) with 4% paraformaldehyde (PFA) for 10 min at room temperature (RT). To detect *cis*-golgi, cells were permeabilized by incubating with 0.1% Triton X-100 for 10 min. After a 60 min in blocking solution (10% goat serum [Gibco], 5% nonfat dry milk, PBS), cells were treated with a mouse anti-GM130 antibody (1/1000, BD Transduction Laboratories) antibody in 10% goat serum and 0.05% Triton X-100 and were again incubated for 1 hr. After three washes, preparations were incubated for 45 min with an Alexa-Fluor-660 conjugated antibody (1:200; Molecular Probes), washed, and mounted in Mowiol (Calbiochem). All procedures were performed at RT. All images were acquired with a Leica TCS SP2 AOBS microscope. Colocalization analysis was performed with ImageJ (National Institutes of Health, Bethesda, MD) following *Sastre et al., 2019*, and the morphometric analysis of mitochondria was performed following *Strack and Usachev, 2017*.

In CD4+ human T lymphocytes, 500 nM MitoTracker (Thermo Fisher Scientific) was used to visualize mitochondria according to manufacturer's instructions. For membrane surface labeling, Wheat Germ Agglutinin-Alexa555 (WGA, Invitrogen) was used. Cells were washed with PBS at 4°C and stained with a dilution of WGA (1/1500) in RPMI supplemented with 30 mM Hepes for 5 min at 4°C. Next, cells were washed twice and fixed with 4% paraformaldehyde for 10 min. Finally, cells were washed and mounted in Mowiol (Calbiochem). Confocal images were acquired with a Zeiss 880 confocal microscope.

## Cell unroofing preparations (CUPs) and Förster resonance energy transfer (FRET)

CUPs were obtained via osmotic shock as previously described (*Oliveras et al., 2020*). Briefly, cells were cooled on ice for 5 min and washed twice with PBS. Next, cells were incubated for 5 min in 1:3 diluted KHMgE (70 mM KCl, 30 mM HEPES, 5 mM MgCl$_2$, 3 mM EGTA, pH 7.5) and were gently washed with nondiluted KHMgE to induce the hypotonic shock. Broken cells were removed from the coverslip by pipetting up and down. After two washes with KHMgE buffer, only membrane sheets remained attached. CUPs were fixed with fresh 4% paraformaldehyde for 10 min at RT and mounted in Mowiol mounting media.

FRET was performed using the acceptor photobleaching configuration. Samples were imaged with a Leica SP2 confocal microscope. Images were acquired before and after YFP bleaching using a 63× oil immersion objective at a zoom setting of 4. Excitation was performed via the 458 and 514

nm lines using an Ar laser, and 465–510 and 525–560 bandpass emission filters were used. FRET efficiency (FRETeff) was calculated using the following equation:

$$(FDafter - FDbefore)/FDbefore \times 100$$

where FDafter: donor fluorescence (Cerulean) after and FDbefore before acceptor (YFP) bleach. Analysis was performed using ImageJ.

## Transmission electron microscopy and correlative transmission electron microscopy

Cells were transfected and, after 24 hr, fixed with 4% PFA and 0.1% glutaraldehyde at RT for 1 hr followed by a treatment with 2% PFA for 30 min. High-pressure freeze cryofixation with liquid $N_2$ and cryosubstitution, Lowicryl resine embedding, polymerization of blocks, and ultrathin sections (60 nm) were performed in collaboration with Unitat de criomicroscòpia electrònica (CCiT, University of Barcelona). Samples were mounted over Formvar-coated grilles, and sections were finally contrasted with uranyl acetate 2% for 15 min. Immunolabeling was performed with the primary antibodies anti-Kv1.3 (Neuromab, 1:30) and anti-Caveolin 1 (1:70, Abcam). Secondary antibodies were conjugated to 12 and 18 nm gold particles as indicated. Samples were imaged using a Tecnai Spirit 120kV microscope. Correlative transmission electron microscopy was performed by the Microscopy Facility of the Department of Biology, University of Padova, as described in *Leanza et al., 2017*.

## Cell death assays, mitochondrial membrane potential measurements, and oxygen consumption rate (OCR) measurements

For the evaluation of apoptosis, plated cells were treated for 18 or 24 hr with the indicated drugs (0.5 µM staurosporine; 10 µM ceramide; 5 µM etoposide) in DMEM without serum and phenol red. After treatment, cells were washed with PBS and suspended in FACS buffer (10 mM HEPES, 140 Mm NaCl, 2.5 mM $CaCl_2$, pH 7.4) containing Annexin V APC and DAPI for 15 min in the dark. Samples were immediately analyzed using either a Gallios flow cytometer (HEK 293 cells) or a microscope (B16F10 cells). To measure mitochondrial membrane potential, cells were incubated with 20 nM tetramethyl rhodamine methyl ester (TMRM) in DMEM without serum and phenol red. Next, samples were diluted up to 5 nM TMRM with more DMEM and analyzed by flow cytometry (FACS-Canto II, Becton Dickinson).

Respiration was measured using an XF24 Extracellular Flux Analyzer (Seahorse, Bioscience). HEK 293 cells, with >60% transfection efficiency, were seeded at $1.5 \times 10^4$ cells/well in 100 µl of DMEM. After 24 hr, the medium was replaced with 670 µl/well of high-glucose DMEM without serum and sodium bicarbonate and supplemented with 10 mM sodium pyruvate and 2 mM L-glutamine. The OCR was measured upon the addition of oligomycin to block ATP synthase (2 µg/ml), FCCP uncoupler (200 nM), and Antimycin A to inhibit complex III (1 µM). All chemicals were added to 70 µl of DMEM. Positive YFP fluorescence was used to monitor cell transfection before OCR measurements by using a Leica microscope (not shown).

## Molecular modeling

Kv1.3 was modeled using high-resolution templates of remote or close homologs available from the Protein Data Bank (PDB; http://www.rcsb.org/pdb) as previously described (*Martínez-Mármol et al., 2013*; *Solé et al., 2019*). Transmembrane domains and the N-terminus (except for the first 49 amino acids [aa]) were modeled with the Kv1.2 potassium channel (PDB code 2R9R). The C terminus and the remaining 49 aa from the N-terminus were modeled with 3HGF (nucleotide-binding domain of the reticulocyte-binding protein Py235) and 1PXE (zinc-binding domain from neural zinc finger factor-1) structures, respectively. The procedure was defined by the i-Tasser online server (http://zhan-glab.ccmb.med.umich.edu/I-TASSER/), and sequence alignments were executed using CLUSTALW from the European Bioinformatics Institute site (http://www.ebi.ac.uk). The homology modeling was performed using the Swiss-Model Protein Modeling Server on the ExPASy Molecular Biology website (http://kr.expasy.org/) under the *Project Mode*. The final molecular graphic representations were created using PyMOL v1.4.1 (http://www.pymol.org/) (*Martínez-Mármol et al., 2013*; *Solé et al., 2019*).

## Oocyte preparation, microinjection, and electrophysiological recordings

Animal handling was carried out in accordance with the guidelines for the care and use of experimental animals adopted by the E.U (RD214/1997). Adult female *Xenopus laevis* (Harlan Interfauna Ibérica) were immersed in cold 0.17% ethyl 3-aminobenzoate methanesulfonate for 20 min, and a piece of ovary was drawn out aseptically. Fully grown immature oocytes, stages V and VI, were isolated and their surrounding layers were removed manually. Cells were kept at 15–16°C in a modified Barth's solution (88 mM NaCl, 1 mM KCl, 2.40 mM $NaHCO_3$, 0.33 mM $Ca(NO_3)_2$, 0.41 mM $CaCl_2$, 0.82 mM $MgSO_4$, 10 mM HEPES [pH 7.4], 100 U/ml penicillin, and 0.1 mg/ml streptomycin) until use. Oocytes were microinjected with 100 nL of cRNA from Kv1.3 WT or Kv1.3CBD$_{less}$ pCDNA3.

Membrane current recordings were performed at 21–25°C, 16–72 hr after injection using a high-compliance two-microelectrode voltage-clamp system (TurboTEC-10CD npi, Tamm). The recording methodology has been described in detail elsewhere (*Morales et al., 1995*; *Olivera-Bravo et al., 2007*). Briefly, oocytes were placed in a 150 µL recording chamber and continuously superfused with normal frog Ringer's solution (115 mM NaCl, 2 mM KCl, 1.8 mM $CaCl_2$, 5 mM HEPES, pH 7.0). The membrane potential was held at $-100$ mV and 20 mV depolarizing voltage steps were applied to +40 mV for a duration of 2.5 s. Membrane currents were low-pass filtered at 30–200 Hz and recorded on a PC, after sampling (Digidata 1200, Molecular Devices, San Jose, CA) at fivefold the filter frequency, using the WCP v.3.2.8 package developed by J. Dempster (Strathclyde Electrophysiology Software, University of Strathclyde, UK).

## Statistics

The results are expressed as the mean ± SE. Student's t-test, one-way ANOVA, and Tukey's post hoc test and two-way ANOVA were used for statistical analysis (GraphPad PRISM v5.01). $p < 0.05$ was considered statistically significant.

## Acknowledgements

Supported by the Ministerio de Ciencia e Innovación (MICINN), Spain (BFU2017-87104-R and PID2020-112647RB-I00 to AF; CSD2008-00005 to AM), the Italian Association for Cancer Research (AIRC IG grant 20286 to IS), the Italian Ministry of University and Education (PRIN 20174TB8KW_004 to IS), the Italian Association for Multiple Sclerosis (to IS), and the European Regional Development Fund. Italian and Spanish laboratories share equal co-responsibility of the work. JC and MPV contributed equally. RP and MNP contributed equally. JC, MPV, and MNP hold fellowships from the Fundación Tatiana Pérez de Guzmán el Bueno and MICINN, respectively. Authors thank to Prof. G Fernández-Ballester (University Miguel Hernández) for his help with the molecular model and Prof. C Deutsch (University of Pennsylvania) for useful discussion. The English editorial assistance of the American Journal Experts is also acknowledged.

## Additional information

### Funding

| Funder | Grant reference number | Author |
| --- | --- | --- |
| Ministerio de Ciencia, Innovación y Universidades | BFU2017-87104-R | Antonio Felipe |
| Ministerio de Ciencia, Innovación y Universidades | PID2020-112647RB-I00 | Antonio Felipe |
| Ministerio de Ciencia, Innovación y Universidades | CSD2008-00005 | Andrés Morales |
| Italian Association for Cancer Research | 20286 | Ildiko Szabó |
| Ministero dell'Istruzione, dell'Università e della Ricerca | PRIN 20174TB8KW_004 | Ildiko Szabó |
| Associazione Italiana Sclerosi Multipla | | Ildiko Szabó |

| European Regional Development Fund | Antonio Felipe |
| --- | --- |

The funders had no role in study design, data collection and interpretation, or the decision to submit the work for publication.

## Author contributions
Jesusa Capera, Formal analysis, Validation, Investigation, Visualization, Methodology, Writing - original draft, Writing - review and editing; Mireia Pérez-Verdaguer, Conceptualization, Formal analysis, Validation, Investigation, Visualization, Methodology, Writing - original draft, Writing - review and editing; Roberta Peruzzo, Juan Martínez-Pinna, Armando Alberola-Die, Formal analysis, Investigation, Methodology; María Navarro-Pérez, Formal analysis, Validation, Investigation, Visualization, Methodology, Writing - review and editing; Andrés Morales, Formal analysis, Funding acquisition, Validation, Investigation, Methodology; Luigi Leanza, Formal analysis, Supervision, Validation, Investigation, Methodology; Ildiko Szabó, Conceptualization, Supervision, Funding acquisition, Validation, Methodology, Writing - original draft, Project administration, Writing - review and editing; Antonio Felipe, Conceptualization, Supervision, Funding acquisition, Validation, Writing - original draft, Project administration, Writing - review and editing

## Author ORCIDs
Jesusa Capera (iD) https://orcid.org/0000-0002-8123-7725
Roberta Peruzzo (iD) https://orcid.org/0000-0001-9209-9068
María Navarro-Pérez (iD) https://orcid.org/0000-0001-8106-9787
Armando Alberola-Die (iD) http://orcid.org/0000-0001-5391-5739
Antonio Felipe (iD) https://orcid.org/0000-0002-7294-6431

## Ethics
Human subjects: The protocol was reviewed and approved by the Ethics Committee of the Universitat de Barcelona and the Banc de Sang i Teixits de Catalunya (BST). Institutional Review Board (IRB00003099). All procedures followed the rules of the Declaration of Helsinki Guidelines. All donors signed a written informed consent and samples were totally anonymous and untraceable.
Animal experimentation: Animal handling was carried out in accordance with the guidelines for the care and use of experimental animals adopted by the E.U (RD214/1997).

## Decision letter and Author response
Decision letter https://doi.org/10.7554/eLife.69099.sa1
Author response https://doi.org/10.7554/eLife.69099.sa2

## Additional files

### Supplementary files
• Transparent reporting form

### Data availability
All data generated or analysed during this study are publicly available on Dryad at https://doi.org/10.5061/dryad.mcvdnck13.

The following dataset was generated:

| Author(s) | Year | Dataset title | Dataset URL | Database and Identifier |
| --- | --- | --- | --- | --- |
| Felipe | 2021 | Data from: A novel mitochondrial Kv1.3-caveolin axis controls cell survival and apoptosis | https://doi.org/10.5061/dryad.mcvdnck13 | Dryad Digital Repository, 10.5061/dryad.mcvdnck13 |

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
