## [Decision Letter]

**Acceptance summary:**

Kv 1.3 channels either cause cell proliferation or apoptosis depending on their sub-cellular localization. In this study, Capera et al. show that the association of Kv 1.3 channels with caveolin is critical for localization to plasma membrane and that the disruption of this interaction causes mitochondrial targeting and triggers apoptosis.

**Decision letter after peer review:**

[Editors’ note: the authors submitted for reconsideration following the decision after peer review. What follows is the decision letter after the first round of review.]

Thank you for submitting your work entitled "A novel mitochondrial Kv1.3-caveolin axis controls cell survival and apoptosis" for consideration by *eLife*. Your article has been reviewed by 3 peer reviewers, and the evaluation has been overseen by a Reviewing Editor and a Senior Editor. The reviewers have opted to remain anonymous.

Our decision has been reached after consultation between the reviewers. Based on these discussions and the individual reviews below, we regret to inform you that your work will not be considered further for publication in *eLife*.

Kv 1.3 channels are involved in cell proliferation of leukocytes and are also known to promote apoptosis in tumor cells. To understand the mechanisms that contribute to these seemingly opposing roles of Kv1.3, this study examines the contribution of interaction between caveolin 1 (Cav1) and Kv1.3 channels on subcellular distribution and functional properties. Multiple lines of evidence suggest that the interaction between Kv1.3 and Cav1 in mitochondria prevents Kv1.3 mediated cell-death. However, the reviewers have raised a number of substantive concerns. Although many of these concerns can be addressed by adding new experiments, it seems unlikely that the revised manuscript can be resubmitted within the next couple of months. We are happy to give due consideration to the revised version of this manuscript if you decide to submit it to *eLife* again but it will be treated as new submission. Please do not hesitate to contact us if you have questions.

*Reviewer #1:*

This manuscript described the effects of the ablation of the Caveolin 1 (Cav1) interaction site in Kv1.3. on subcellular distribution, functional properties and effects on apoptosis. The main conclusion of the paper is that the interaction between Kv1.3 and Cav1 plays a crucial role not only in the proper plasma membrane localization of Kv1.3, resulting in an enrichment of mitochondrial localization, but that the interaction with Cav1 at the mitochondria also modulates the induction of apoptosis. Such an effect would justify the proposal of a Cav1-Kv1.3 axis, and would be a very relevant finding adding to the emerging importance of both Cav1 and Kv1.3 in physiology and pathology, especially in cancer.

The main concern is the possibility that an "overload" of the mitochondria with either wild type or CBDless Kv1.3 is responsible for the observed effects.

1. Whereas the enrichment in mitochondrial Kv1.3 in CBDless channels leaves little doubt, to what extent that interaction at the mitochondria is important for apoptosis is less clear. Kv1.3 wild type overexpression induces an increase of apoptosis on its own. If Cav1 binding-defective mutants are more abundant at the mitochondria, this alone could explain the increase in apoptosis described. The effects reported in both Jurkat and 3T3-L1 could in principle be also explained if the excess of Kv1.3 is equally targeted to mitochondria in Jurkat cells regardless of CBD due to the low Cav1 level, and if knockdown of Cav1 reduces PM Kv1.3 in preadipocytes and therefore increases mitoKv1.3 (at least it seems to alter the ratio between mature and non-glycosylated channel, Figure 7F).

2. As the authors state, it appears that the reduced PM activity of Kv1.3CBDless is due to trafficking defects (less targeting to the membrane and shorter half-life). Actually, the mutant is activated at less depolarized potentials, suggesting a more favorable transition to an open state. Inhibition of mitoKv1.3 by Bax or pharmacological agents triggers apoptosis in several cell types. It is therefore counterintuitive, or at least it is not directly explained in the manuscript, that overexpression of a variant of the channel, which is in fact "more active" than the wild type induces more apoptosis, unless the effect is different from the physiological role of mitoKv1.3.

3. It would be extremely interesting to know what the effect of mitochondria-targeted Kv1.3 inhibitors like the ones described previously by some of the authors have on cells overexpressing Kv1.3 (wild type and CBDless), and if such agents can counteract the effect of Cav1 knockdown. One would expect that such treatment would be protective rather than apoptosis-inducing.

*Reviewer #2:*

The comparison of WT-Kv1.3 and the CBDless Kv1.3 mutant (166-FQRQVWLLF-174 to 166-AQRQVGLLA-174) is the central focus of this paper. In an earlier paper (Sci Rep 2016;6:22453), the authors reported that the CBDless Kv1.3 mutant did not interact with caveolin-1 and exhibited reduced lipid raft portioning. In the present paper, the authors report that the CBDless Kv1.3 mutant preferentially accumulates in mitochondria, resulting in altered mitochondrial physiology and cell survival.

CBDless Kv1.3 mutant

• In *Xenopus oocytes*, the CBDless Kv1.3 mutant produced roughly 1/4th the current amplitude of WT Kv1.3. The authors conclude that the reduced current density is the consequence of reduced surface abundance. However, reduced current density could also be due to decreased production of the CBDless Kv1.3 mutant. A Western blot comparing Kv1.3 protein expression in *Xenopus oocytes* injected with WT Kv1.3 versus the CBDless mutant could exclude this possibility.

• In *Xenopus oocytes*, the CBDless triple mutation in the N-terminus alters voltage-dependence of activation and accelerates C-type inactivation (which involves changes in the outermost selectivity filter; J Gen Physiol 141: 151-160; Nat Struct Mol Biol. 2017; 24: 857-865). Clearly, the CBDless mutation impacts regions of Kv1.3 distant from the N-terminus. A more detailed electrophysiological characterization of the CBDless mutant channel is warranted.

• Since many studies are done on HEK293, B16F10 melanoma or Jurkat cells transfected with WT Kv1.3 or the CBDless Kv1.3 mutant, it is essential to show electrophysiology data for both channels in these cells. Mutations of 1 or 2 of the three aromatic residues in the CBD may be sufficient to reduce caveolin-1 binding and be less disruptive to the channel. Point mutations in the CBD of HCN4 are sufficient to alter caveolin-1 binding (J Mol Cell Cardiol. 2012;53:187-95).

• If Kv1.3's association with caveolin-1 is essential to localize Kv1.3 in lipid raft membrane microdomains in the plasma membrane, how do Kv1.3 channels in caveolin-1 deficient Jurkat cells translocate to the plasma membrane and generate normal Kv1.3 currents (J Biol Chem. 2001;276:12249-56)? In Jurkat cells transfected with WT Kv1.3 versus CBDless Kv1.3 channels, is the Kv1.3 membrane-to-mitochondrial ratio, mitochondrial morphology and mitochondrial function different?

MitoKv1.3, mitochondrial dysfunction, apoptosis

• In Figure 3, please show membrane Kv1.3 expression normalized to Na-K-ATPase in cells transfected with WT Kv1.3 or CBDless Kv1.3. This could be correlated with Kv1.3 channel density determined electrophysiologically.

• In Figure 6C, one Kv1.3-spot is shown in HEK293 cells transfected with WT Kv1.3, and in Figure 6E, one Kv1.3-spot is shown in HEK293 cells transfected with CBDless Kv1.3. Wouldn't you expect more Kv1.3 channels spots in the mitochondria? In Figure 7C, Kv1.3 is shown inside the organelle and not at the inner mitochondrial membrane. Do you have any images showing Kv1.3 and caveolin-1 in the inner mitochondrial membrane?

• If caveolin-1's interaction with mitoKv1.3 alleviates the pro-apoptotic activity of mitochondrial Kv1.3, caveolin-1 deficient Jurkat cells (that contain mitoKv1.3 unbound to caveolin-1) should be more prone to apoptosis than caveolin-1 and Kv1.3 containing cells, and over-expression of caveolin-1 in Jurkat cells should suppress apoptosis. This might be worth testing.

• Is increased apoptosis in 3T3-L1 cells following caveolin-1 knockdown due to the pro-apoptotic effects of mitoKv1.3 (as you suggest), or is it due to p53-p21-dependent induction of mitochondrial dysfunction and cellular senescence caused by caveolin-1 deficiency (Aging Cell 2017;16:773-784)? One could distinguish between these possibilities by over-expressing Kv1.3 in 3T3-L1 cells such that there is an excess of mitoKv1.3 over caveolin-1. If apoptosis is enhanced, it would suggest that mitoKv1.3 is a driver of apoptosis.

• Human T cells express caveolin-1, and caveolin-1 deficiency in T cells reduces effector function (J Immunol. 2017; 199: 874-884). Based on your results, caveolin-1 deficient primary human T cells would be predicted to express less surface Kv1.3 and more mitoKv1.3, exhibit altered mitochondrial morphology and physiology, and be more susceptible to apoptosis. This might be worth testing.

• Accumulation of Kv1.3 in the ER and Golgi when CBD is mutated (Figure 2D) may affect the physiology of these organelles. The authors may wish to address/discuss this.

*Reviewer #3:*

Capera et al. cover an interesting topic in their manuscript, which reports a novel role of the interaction between caveolin 1 and Kv1.3 channels. The Authors present a detailed analysis on the possible function of caveolin and Kv1.3 in cellular processes such as apoptosis. They revealed that mitochondrion associated Kv1.3 without caveolin 1 binding domain is pro-apoptotic, WT Kv1.3 show colocalization with caveolin in the mitochondrial inner membrane, contributes to the mitochondrial dysfunction via destructing the structure of cisterna. Authors concluded that Kv1.3-caveolin 1 "crosstalk" can sensitize the cancer cells to apoptosis. Based on these my questions and comments are the following:

1. Authors in a series of experiments use Cav1 knock-down HEK cell line. However, I did not find in the manuscript if they applied it in the mitochondrial function and apoptosis experiments. As all the experiments related to Cav1-driven Kv1.3 targeting I suppose it would be practical to use a cell line lacking endogenous Cav1 and then "re-add" Cav1 with transfection of Cav1 plasmid e.g. for organelle-localization experiments, mitochondrial location, apoptosis.

2. For electrophysiology Authors chose oocytes instead of HEK cells. Why did not the authors show data which were measured in HEK cells? In their previous paper the authors demonstrated that in Cav1- HEK cells the Kv1.3 channel had slower inactivation kinetics than in those with Cav1 expression, and not a drastic reduction in the current happened upon Cav1 knock-down. Here the data show that prevention of Cav1 binding to Kv1.3 channels results in a drastic change. I think these outcomes, at least in part, are contradictory. Do the Authors have any explanation for that? Is it possible that mutation eliminated the interaction between the channel and another (not Cav1) protein necessary for PM targeting?

3. For the TEM images the authors mentioned that Kv1.3 channels are localized to mitochondria detected using the immunogold labeling technique. I could observe only one-two gold beads/mitochondria (or organelles), which I suppose is unexpectedly low as Kv1.3 protein was overexpressed (and also large fraction of channels target to mitochondria). On the other hand, I wonder if any specific markers for organelles were applied to identify them.

[Editors’ note: further revisions were suggested prior to acceptance, as described below.]

Thank you for resubmitting your work entitled "A novel mitochondrial Kv1.3-caveolin axis controls cell survival and apoptosis" for further consideration by *eLife*. Your revised article has been evaluated by Kenton Swartz (Senior Editor) and a Reviewing Editor.

The manuscript has been improved but there are some remaining issues that need to be addressed, as outlined below:

1. One of the claims made in the paper: " interactions between caveolin and Kv1.3 are essential for apoptosis" seems slightly overstated. We recommend the authors to soften this claim and discuss alternative possibilities. Their work clearly provides evidence for improper trafficking of Kv1.3 CBDless to the plasma membrane and ectopic expression in mitochondria, resulting in apoptosis. They have also shown that caveolin is involved in modulating apoptosis. Previous work from their lab has shown caveolin interacts with Kv1.3 and increases C-type inactivation and fails to interact with CBDlessKV1.3. It remains unclear whether there is any direct interaction/cross-modulation between Kv1.3 and caveolin in the mitochondria, causing apoptosis, and that requires more direct evidence to support the claim.

There is no doubt that both caveolin and Kv1.3 are involved in apoptosis and apoptosis IS dependent on Kv1.3-caveolin axis, but the direct interaction between these two proteins in the mitochondria warrants further investigations. The experimental evidence provided in the manuscript are suggestive of such an interaction in mitochondria and resulting apoptosis, but it is not definitive. Addressing it in the discussion will benefit the readers.

2. Regarding the electrophysiological properties of the mutant channel, against the view of the authors and even though the construct appears toxic to mammalian cells and the experiments need to be done in oocytes, if the mutant activates „earlier", it is in our view more active at any potential. Also, in oocytes, if the transport to the membrane is altered, then it would be expected that the currents are smaller. A decrease in the input resistance of the oocyte would also rather indicate a "higher" basal activity. If we understand correctly, the role of Kv1.3 in apoptosis relies on the fact that reduction of its activity through bax blockade triggers ∆ψm hyperpolarization and then apoptosis, so it is counterintuitive how a "more active" channel is proapoptotic. Importantly, how activation of Kv1.3 happens with the values of ∆ψm in the context of mitochondria (140 mV negative to the activation potential of Kv1.3) remains an unanswered question. We think that this issue merits a few words in the discussion. Moreover, Cav1 maintains mitochondrial depolarization, as its knockdown induces hyperpolarization. This is against what one would expect if it acts partly through Kv1.3. The concept "Kv1.3-mediated cell death" is somewhat misleading if Kv1.3 blockade is what triggers apoptosis. Why does then mitoKv1.3 overexpression promote apoptosis? We might well have got the point wrong, but we think other readers would have the same problem.

3. Disregarding the fact that Kv1.3 overexpression on its own reduces respiration by almost one-half (Figure 5), the mitochondrial fragmentation induced by Kv1.3CBDless deserves some comments in this context.

*Reviewer #1:*

This manuscript addresses the importance of Cav1 interaction for Kv1.3 targeting to the plasma membrane and the relevance of the interaction for the impact of Kv1.3 in apoptosis. In this revised version, the authors have provided further evidence of the specificity of the effect. Nevertheless, I have still some substantial questions that the authors probably can address by changing the wording of their statements.

Regarding the electrophysiological properties of the mutant channel, against the view of the authors and even though the construct appears toxic to mammalian cells and the experiments need to be done in oocytes, if the mutant activates „earlier", it is in my view more active at any potential. Also, in oocytes, if the transport to the membrane is altered, then it would be expected that the currents are smaller. A decrease in the input resistance of the oocyte would also rather indicate a "higher" basal activity. If I understand correctly, the role of Kv1.3 in apoptosis relies on the fact that reduction of its activity through bax blockade triggers ∆ψm hyperpolarization and then apoptosis, so it is counterintuitive how a "more active" channel is proapoptotic. Importantly, how activation of Kv1.3 happens with the values of ∆ψm in the context of mitochondria (140 mV negative to the activation potential of Kv1.3) remains an unanswered question. I think that this issue merits a few words in the discussion. Moreover, Cav1 maintains mitochondrial depolarization, as its knockdown induces hyperpolarization. This is against what one would expect if it acts partly through Kv1.3. The concept "Kv1.3-mediated cell death" is somewhat misleading if Kv1.3 blockade is what triggers apoptosis. Why does then mitoKv1.3 overexpression promote apoptosis? I might well have got the point wrong, but I think other readers would have the same problem.

Disregarding the fact that Kv1.3 overexpression on its own reduces respiration by almost one-half (Figure 5), the mitochondrial fragmentation induced by Kv1.3CBDless deserves some comments in this context.

Figure S6. This is indeed a very nice control. However, it is very hard to assess how efficiently YMVIii is targeted to the mitochondria. Loss of membrane targeting would, in any case, increase Pearsons's correlation against any intracellular marker. It would be much more convincing if the copurification experiments in Figure 3 or 7 would be reproduced here.

*Reviewer #2:*

The authors have presented an elegant work on the role of Kv1.3 in cellular homeostasis. Authors clearly demonstrated that CBDless Kv1.3 failed to localize on the plasma membrane. The work highlights the significance of interactions between Kv1.3 and caveolin in cellular apoptosis. The data presented supports the notion that improper trafficking of Kv1.3 to plasma membrane can result in ectopic expression in the mitochondrial membrane, which can dysregulate protein interactions, causing apoptosis.

In my opinion, the authors have addressed all the reviewers comments and the manuscript is ready to be accepted for publication. No further edits are required. Congratulations to all the authors for an intensive work presented in the manuscript.

This is one minor suggestion to the authors for future work (and is not required for this manuscript): To test the currents in oocytes where caveolin and Kv1.3 are co-expressed and compare it with CBDless clones (with and without the co-expression of caveolin). This might provide more direct evidence into Kv1.3 modulation by caveolin. This, of course, might not address their role pertaining to mitochondrial functions, as oocyte expression will be in the plasma membrane, but will certainly provide insights into functional interactions. Lack of interaction between CBDless Kv1.3 and caveolin in oocytes will provide validation that many reviewers raised. Again, such intensive study is outside the scope of current manuscript and is merely suggested here for future work.

---

## [Author Response]

[Editors’ note: the authors resubmitted a revised version of the paper for consideration. What follows is the authors’ response to the first round of review.]

Reviewer #1:This manuscript described the effects of the ablation of the Caveolin 1 (Cav1) interaction site in Kv1.3. on subcellular distribution, functional properties and effects on apoptosis. The main conclusion of the paper is that the interaction between Kv1.3 and Cav1 plays a crucial role not only in the proper plasma membrane localization of Kv1.3, resulting in an enrichment of mitochondrial localization, but that the interaction with Cav1 at the mitochondria also modulates the induction of apoptosis. Such an effect would justify the proposal of a Cav1-Kv1.3 axis, and would be a very relevant finding adding to the emerging importance of both Cav1 and Kv1.3 in physiology and pathology, especially in cancer.The main concern is the possibility that an "overload" of the mitochondria with either wild type or CBDless Kv1.3 is responsible for the observed effects.

First of all, we apologize for the delay answering the reviewers concerns. The initial and total lockdown in both Spain and Italy plus further partial lockdowns during the last months have made very difficult to address all requests in due time.

Thank you very much for your kind review considering our findings very relevant to physiology and pathophysiology of, especially but not limited to, cancer cells. Indeed one of the most relevant findings has been the identification of the essential role of the mitochondrial Cav/Kv1.3 association controlling cell survival and apoptosis.

Although the overload of either Kv1.3 could exert some observed effects, this is unlikely because Kv1.3 WT sensitizes to apoptotic stimuli rather than generating apoptosis by itself, with no changes in mitochondrial morphology in the presence of caveolin (Figure 4-7). Only the CBD_less_ mutant triggers massive apoptosis (Figure 4 and 6); thereby arguing against the observation of the Reviewer. To further examine this point, we performed new experiments using primary human T-cells and 3T3 fibroblasts, where we modulated only expression of caveolin (but not of Kv1.3). These new data further support our findings (Figure 7-10). Importantly, additional, new data exploiting the YMVIii mutant that, being intracellular retained (Martinez-Marmol et al., 2013; J Cell Sci 126, 5681-5691), partially shares CBD_less_ intracellular localization and also targets to mitochondria, but preserves cav 1 interaction, did not generate apoptosis (Supplemental Figure 6).

1. Whereas the enrichment in mitochondrial Kv1.3 in CBDless channels leaves little doubt, to what extent that interaction at the mitochondria is important for apoptosis is less clear. Kv1.3 wild type overexpression induces an increase of apoptosis on its own.

Kv1.3, as mentioned above, sensitives to apoptosis rather generates apoptosis (see Figure 4-7). The Reviewer is right that in HEK293 cells (see Figure 6A), the % of apoptotic cells is relatively high after expression of WT Kv1.3, but it is significantly less than in the case of cells expressing Kv1.3 CBDless. Please note also that in Suppl. Figure 5, where the B16F10 melanoma cells were first sorted following transfection and then treated with pro-apoptotic stimuli (or left untreated), the % of apoptotic cells is again much higher in the case of the cells expressing the mutant channels. Importantly, even in the HEK293 transfected with Kv1.3 WT, the cells look healthy and mitochondria maintained correct cristae organization (Figures 6 and 7) and the mitochondrial membrane potential as well the maximal respiration are similar to that observed in control cells (Figure 5). In addition, results obtained with the Kv1.3 YMVIii mutant also supports our claims (Supplemental Figure 6).

If Cav1 binding-defective mutants are more abundant at the mitochondria, this alone could explain the increase in apoptosis described.

The channel mitochondrial accumulation is not exclusively responsible for the apoptotic effects. Indeed, YMVIii mutant reroutes to mitochondria but generates no apoptosis (Supplemental Figure 6). In addition, the expression of WT Kv1.3 in T-cells (Jurkat and primary T-lymphocytes, low cav) promotes similar apoptosis than that caused by Kv1.3 CBDless, but the introduction and/or the endogenous expression of Cav partially protects cells from apoptosis in Kv1.3WT but not in CBDless (Figure 7 and 10. The ablation of cav 1 in 3T3 fibroblasts with no alteration of Kv1.3 further supports our data (Figure 7). Therefore, all these data strongly suggest that the apoptosis is mostly related to the absence of cav/Kv1.3 association rather than to the unique expression of Kv1.3 by itself. In fact, caveolin by itself is claimed to be anti-apoptotic (just few examples: Yang et al., Cancer Invest. 2012, 30:453-462; Codenotti et al., Cancer Lett. 2021, 505:1-12; Aberg et al., Apoptosis. 2020, 25:519-534).

The effects reported in both Jurkat and 3T3-L1 could in principle be also explained if the excess of Kv1.3 is equally targeted to mitochondria in Jurkat cells regardless of CBD due to the low Cav1 level, and if knockdown of Cav1 reduces PM Kv1.3 in preadipocytes and therefore increases mitoKv1.3 (at least it seems to alter the ratio between mature and non-glycosylated channel, Figure 7F).

Yes, the reviewer is correct. When Kv1.3 does not associate with cav, the channel impairs membrane expression (lower non-glycosylated band). However, the level of Kv1.3 rerouting to mitochondria is not proportional to the level of apoptosis. In addition, the new YMVIii mutant data, which targets to mitochondria but associates with caveolin, clearly supports that the level of Kv1.3 channel in mitochondria is not the unique cause (supplemental Figure 6). Thus, the mitoKv1.3 association with caveolin rather than the level of mitoKv1.3 by itself should be the responsible of such effect. Furthermore, we have incorporated data with fresh human primary T-cells, which naturally lack expression of Cav. In such cells, both Kv1.3 channels (WT and CBDless) generate similar level of basal apoptosis, but the introduction of external caveolin partially protects the cells when the channel can interact with Cav, i.e. only in Kv1.3 WT (Figure 10). These data therefore further support Jurkat and 3T3-L1 results (Figure 7).

2. As the authors state, it appears that the reduced PM activity of Kv1.3CBDless is due to trafficking defects (less targeting to the membrane and shorter half-life). Actually, the mutant is activated at less depolarized potentials, suggesting a more favorable transition to an open state. Inhibition of mitoKv1.3 by Bax or pharmacological agents triggers apoptosis in several cell types. It is therefore counterintuitive, or at least it is not directly explained in the manuscript, that overexpression of a variant of the channel, which is in fact "more active" than the wild type induces more apoptosis, unless the effect is different from the physiological role of mitoKv1.3.

To our knowledge, the fact that CBDless activates earlier does not mean that is more active. In fact, the current generated by Kv1.3 CBDless is much lower. The electrophysiology was performed in oocytes because the plasma membrane of mammalian cells was severely altered by the Kv1.3 CBDless-related massive apoptosis. Therefore, trying to establish a close relationship between channel kinetics and physiology is misleading. Because we could not perform patch clamp in CBDless-transfected mammalian cells, we analyzed the lipid composition of the PM under apoptosis. CBDless dependent apoptotic cells exhibited a reduction in membrane cholesterol (12%), phosphatidylserine + phosphatidylinositol (10%) and sphingolipids (5%), which severely affects the membrane integrity to be patch-clamped. Seals broke and cells died. Voltage clamp studies in oocytes solely demonstrate a functional channel which generates tetramers despite altering an important motif lying close the Shaker tetramerization domain. In fact, we also saw important effects on the membrane resistance of oocytes (input resistance, Supplemental Figure 3), which further supports the weakness of the membrane in severely apoptotic mammalian cells (see Figure 6D). Our results demonstrate that CBDless-injected oocytes displayed about half membrane resistance than WT (Supplemental Figure 3). These changes in membrane (HEK and oocytes) are concomitant to important losses of membrane integrity during apoptosis.

On our opinion the data presented here do not contradict the well-delineated mechanism by which mitoKv1.3 block triggers apoptosis. We here further decipher a new mechanism involving Kv1.3, which does not argue against the previous one and further reinforces the role of mitoKv1.3 in this important process. The mitoCav, acting as antiapoptotic (as widely demonstrated) regulates the mitoKv1.3-dependent apoptotic effects. The possibility of a competition between the Bax effect and the Cav effect is feasible and would fine-tune the mitoKv1.3-related apoptotic control under different insults, but addressing this point would be far beyond the scope of the present manuscript.

3. It would be extremely interesting to know what the effect of mitochondria-targeted Kv1.3 inhibitors like the ones described previously by some of the authors have on cells overexpressing Kv1.3 (wild type and CBDless), and if such agents can counteract the effect of Cav1 knockdown. One would expect that such treatment would be protective rather than apoptosis-inducing.

We thank the reviewer for this inquire. However, we do not fully understand her/his claims in the context of our work. We humbly think that the reviewer misunderstands the CBDless activity (see above). Anyway, The PAPTP mitoKv1.3 inhibitor has been used as requested. However, the data generates misunderstandings as expected. As previously published, PAPTP (Leanza et al., 2017, Cancer Cell 31, 516–531) triggered apoptosis in the Kv1.3 expressing cells similarly to the intrinsic apoptosis inducer staurosporine. As previously published, cells with low activity of Kv1.3, such as primary human fibroblasts, MSCs and, in our case, with CBDless channel, PAPTP has little effects. As abovementioned, we claim that the CBDless channel is not more active. Basal apoptosis in the CBDless-expressing cells was extremely high; thereby, the slightly increase of apoptosis triggered by PAPTP, although logical, is difficult to interpret. We do not understand why PAPTP should counteract the apoptosis in CBDless. Furthermore, the absence of caveolin-Kv1.3 association would further increase apoptosis because caveolin should be contemplated as anti-apoptotic in many cell models.

The effectivity of mitoKv1.3 inhibitors, such as PAPTP, might be influenced by different factors. Thus, (i) the activity and expression of Kv1.3 channels (Arcangeli et al., 2009; Leanza et al., 2013); (ii) cells with elevated Kv1.3 exhibit a hyperpolarized IMM (Hockenbery, 2010); (iii) redox state (e.g., Sabharwal and Schumacker, 2014) and (iv) membrane permeability to molecules (such as PAPTP) increase oxidative stress above a critical threshold triggering apoptosis. In fact, loss of *mitochondrial* membrane potential (ΔΨ_m_) has been shown to be an early event during *apoptosis* in some systems. Thus, we show that both channels (WT and CBDless) triggered hyperpolarization in both HEK 293 and oocytes (Figure 5 and Suppl Figure 3).

The main issue of our manuscript deals with the association of Kv1.3 with caveolin rather than the Kv1.3CBDless activity by itself. Therefore, many uncertain inputs, such as changes in membrane lipids, massive apoptosis, minor membrane resistance and more permeable plasma membrane in CBDless channel, indicate that raising conclusions about blocking channel activity should be discarded. However, our data is in accordance with what it has been described before – PAPTP triggers apoptosis – and the massive apoptosis generated by the CBDless channel slightly increased nor decreased by PAPTP. In fact as abovementioned, the CBDless channel is not more active; thereby, the data is much debatable. In this scenario, although we include this data in Author response image 1 for the internal report evaluation, the inclusion of this uncertain data in the main body of the manuscript would be unclear. We humble demand not to be included in the manuscript because would raise confusion.

Reviewer #2:The comparison of WT-Kv1.3 and the CBDless Kv1.3 mutant (166-FQRQVWLLF-174 to 166-AQRQVGLLA-174) is the central focus of this paper. In an earlier paper (Sci Rep 2016;6:22453), the authors reported that the CBDless Kv1.3 mutant did not interact with caveolin-1 and exhibited reduced lipid raft portioning. In the present paper, the authors report that the CBDless Kv1.3 mutant preferentially accumulates in mitochondria, resulting in altered mitochondrial physiology and cell survival.

First of all, we apologize for the delay answering the reviewers concerns. The initial and total lockdown in both Spain and Italy plus further partial lockdowns during the last months have made very difficult to address all requests in due time.

Thank you very much for your kind report. We agree that this is indeed one message in the paper, which, however additionally highlight for the first time that mitoKv1.3 association to caveolin is able to fine-tune cell survival and apoptosis. Caveolin has been claimed as antiapoptotic protector (just few examples: Yang et al., Cancer Invest. 2012, 30:453-462; Codenotti et al., Cancer Lett. 2021, 505:1-12; Aberg et al., Apoptosis. 2020, 25:519-534) but we here show that it functions by specific associations with a novel target, mitoKv1.3.

CBDless Kv1.3 mutant• In *Xenopus* oocytes, the CBDless Kv1.3 mutant produced roughly 1/4th the current amplitude of WT Kv1.3. The authors conclude that the reduced current density is the consequence of reduced surface abundance. However, reduced current density could also be due to decreased production of the CBDless Kv1.3 mutant. A Western blot comparing Kv1.3 protein expression in *Xenopus* oocytes injected with WT Kv1.3 versus the CBDless mutant could exclude this possibility.

The reviewer is correct. Although our data indicated that the main reason triggering lower activity is the intracellular retention, following reviewer’s suggestion, we now include a Western blot comparing Kv1.3 protein expression in *Xenopus oocytes* injected with WT Kv1.3 versus the CBDless mutant (Supplemental Figure 3A). Protein expression demonstrated that the level of protein in CBDless-injected oocytes is similar to that of the WT channel.

• In Xenopus oocytes, the CBDless triple mutation in the N-terminus alters voltage-dependence of activation and accelerates C-type inactivation (which involves changes in the outermost selectivity filter; J Gen Physiol 141: 151-160; Nat Struct Mol Biol. 2017; 24: 857-865). Clearly, the CBDless mutation impacts regions of Kv1.3 distant from the N-terminus. A more detailed electrophysiological characterization of the CBDless mutant channel is warranted.

While we acknowledge the useful suggestion of the Reviewer, we feel that being the main aim of the present work that of deciphering whether the association of Kv1.3 and caveolin influences cell survival and apoptosis, the electrophysiological characterization of the CBDless mutant is not strictly related to this story. Nonetheless, as requested by the reviewer, we introduced into this version new biophysical data from the CBDless mutant channel (Supplemental Figure 3). This include membrane potential, membrane resistance, conductance and cumulative inactivation analysis. The rationale for including a partial electrophysiological characterization of the CBDless mutant was the fact that we were concerned about the possibility that modifying the CBD, next to the tetramerization domain, could impair the oligomerization of the channel. In our previous version, we analyzed few parameters to demonstrate that the CBDless was functional in terms of tetramerization and conductivity. The effects for a limited caveolin interaction were addressed previously as you mention above (Sci Rep 2016;6:22453). In fact, Kv1.3CBDless-exressing cells undergo severe apoptosis and changes in membrane lipid composition and membrane integrity are evident (Figure 6). Because of that, patch-clamp was impossible to achieve in mammalian culture cells. We analyzed the lipid composition of Kv1.3CBD-less HEK cell membrane and they exhibited about 12% cholesterol, 10% phosphatidylserine + phosphatidylinositol and 5% sphingolipids less in their membranes than Kv1.3 WT cells. This clearly will affect the integrity of the membrane as well as the kinetic of channels (Cholesterol regulation of Ion channels and receptors, I. Levitan and F.J. Barrantes eds. 2012. DOI: 10.1002/9781118342312. John Wiley and Sons, Inc). The data is concomitant with a 50% decrease in membrane input resistance of Kv1.3 CBDless-injected oocytes. In this scenario, with such amount of inputs affecting electrophysiological parameters, we honestly think that raising conclusions from those results would be extremely uncertain. Indeed, we mention when describing the newly introduced biophysical parameters that any biophysical conclusion from this CBDless mutant should be taken with caution.

• Since many studies are done on HEK293, B16F10 melanoma or Jurkat cells transfected with WT Kv1.3 or the CBDless Kv1.3 mutant, it is essential to show electrophysiology data for both channels in these cells.

We initially desired to perform the electrophysiology in our mammalian cell models rather than in oocytes. However, as abovementioned, Kv1.3CBDless triggers important changes at the cell physiology and membrane integrity. First, membrane cholesterol and sphingolipids decreased, triggering membrane weakness impeding patch clamp experiments. Therefore, seals were extremely difficult to achieve and cells died just applying the whole cell configuration. Second, severe apoptotic cells exhibit an extremely sensitive cell membrane integrity (see input resistance oocytes, Supplemental Figure 3). This is clearly observed in electron micrographs throughout the manuscript (i.e. Figure 6). We do not know whether the former or the last goes first but, this clearly impeded the study. Finally, we chose oocytes because, under these circumstances, they exhibit an extremely resistant membrane for the validation of channel tetramerization and functionality. To sum up, mammalian cell models were not suitable for electrophysiology with the CBDless mutant.

Mutations of 1 or 2 of the three aromatic residues in the CBD may be sufficient to reduce caveolin-1 binding and be less disruptive to the channel. Point mutations in the CBD of HCN4 are sufficient to alter caveolin-1 binding (J Mol Cell Cardiol. 2012;53:187-95).

Although the reviewer is correct regarding HCN4, in our case what matters is the disruption of caveolin association with Kv1.3 rather than a disruption of the channel activity (or protein). In fact, the CBDless channel tetramerizes (Supplemental Figure 2), is functional (Supplemental Figure 3) and associates to Kvβ subunits (see Author response image 2). That interacts with Kv1 channels at a site that lies next to the CBD.

**Author response image 2. respfig2:** 

Consensus sequences for the caveolin binding domains are: ΦXΦXXXXΦ,ΦXXXXΦXXΦ, or ΦXΦXXXXΦXXΦ, where Φ = Trp, Phe, or Tyr (Couet et al., J Biol Chem. 1997;272(10):6525-33). As suggested by the reviewer, we have performed the experiments, which include mutations of 1 or 2 residues (new Supplemental Figure 1). Only when the second cluster of aromatic residues was substituted, the association to caveolin is impaired. However, both clusters seem to cooperatively participate in the Kv1.3/cav colocalization. We believe that although the second cluster mutant triggers minor caveolin/Kv1.3 association, the data do not alter the significance of our claims.

• If Kv1.3's association with caveolin-1 is essential to localize Kv1.3 in lipid raft membrane microdomains in the plasma membrane, how do Kv1.3 channels in caveolin-1 deficient Jurkat cells translocate to the plasma membrane and generate normal Kv1.3 currents (J Biol Chem. 2001;276:12249-56)?

The reviewer is correct and this issue opens an exciting debate. This is one of our current laboratory goals. Our working hypothesis is that in cells, with a low expression of caveolin, other post-translational mechanisms or protein interactions could take the lead. We are currently deciphering the role of these mechanisms on the Kv1.3-dependent physiology in lymphocytes but addressing this issue in the present manuscript is beyond the scope in our opinion.

In Jurkat cells transfected with WT Kv1.3 versus CBDless Kv1.3 channels, is the Kv1.3 membrane-to-mitochondrial ratio, mitochondrial morphology and mitochondrial function different?

This is a very relevant question. In addition, some additional inquiries from the editorial report demanded experiments in primary human T lymphocytes. Therefore, to concentrate our efforts, in order to answer this query, we preferred to use freshly isolated human T lymphocytes rather than a cell line. These, more physiological data, shows that in fresh blood human T-lymphocytes, which do not express caveolin, Kv1.3 WT and Kv1.3CBD_less_ channels triggered similar effects. Thus, the membrane and mitochondrial colocalization (confocal studies, Figure 8), the mitochondrial morphology (confocal studies, Figure 9) and the mitochondrial function by tetramethylrhodamine, methyl ester (TMRM) dye (mitochondrial membrane potential, Figure 10) were similar with both channels. In addition, annexin V (apoptosis) showed that both channels triggered similar cell apoptosis but, when caveolin was introduced in T-lymphocytes, apoptosis was partially counteracted in cells expressing Kv1.3 WT (Figure 10 B). Furthermore, new data with a selected Jurkat cell line that endogenously expresses caveolin (Figure 7) further confirms that the presence of caveolin triggers certain protection against apoptosis with Kv1.3WT but not with CBDless.

MitoKv1.3, mitochondrial dysfunction, apoptosis• In Figure 3, please show membrane Kv1.3 expression normalized to Na-K-ATPase in cells transfected with WT Kv1.3 or CBDless Kv1.3.

We thank the reviewer for this suggestion. A new panel in Figure 3B has been incorporated as required.

This could be correlated with Kv1.3 channel density determined electrophysiologically.

We humbly apologize, but this correlation cannot be obtained because it was impossible to perform electrophysiology on mammalian cells expressing Kv1.3 CBD_less_ (see above).

• In Figure 6C, one Kv1.3-spot is shown in HEK293 cells transfected with WT Kv1.3, and in Figure 6E, one Kv1.3-spot is shown in HEK293 cells transfected with CBDless Kv1.3. Wouldn't you expect more Kv1.3 channels spots in the mitochondria?

The Reviewer is correct, however in the same figure (H), there are two spots in mitochondria of HEK293 cells transfected with CBD-less Kv1.3. Figure 7C also shows two spots for Kv1.3 and only one spot for Cav, while both proteins are well-detectable in purified mitochondria by Western blot. We have to take into consideration that in the TEM images, using a very limited dilution of antibody, as we did use, we assure that the staining is specific. More staining often represents increase in background, false labeling and poor structural preservation.

In Figure 7C, Kv1.3 is shown inside the organelle and not at the inner mitochondrial membrane. Do you have any images showing Kv1.3 and caveolin-1 in the inner mitochondrial membrane?

New images have been added as requested.

• If caveolin-1's interaction with mitoKv1.3 alleviates the pro-apoptotic activity of mitochondrial Kv1.3, caveolin-1 deficient Jurkat cells (that contain mitoKv1.3 unbound to caveolin-1) should be more prone to apoptosis than caveolin-1 and Kv1.3 containing cells, and over-expression of caveolin-1 in Jurkat cells should suppress apoptosis. This might be worth testing,

We want to thank the reviewer for this inquiry. Evidence in the literature suggest that some leukocytes, that do not initially express caveolin, may express the protein under certain states of activation (Hatanaka et al., Biochem Biophys Res Commun 1998, 253: 382387). Indeed, we managed to isolate some Jurkat cells that started to express endogenously caveolin (most probably due to several freezing/thawing cycles they underwent due to the lock down). New data (Figure 7 D, E) with these Jurkat cav+ cells have been incorporated. As suggested by the reviewer, in Jurkat cells, which do express caveolin endogenously, the apoptosis was partially counteracted. However, it must be said that regular, primary Tlymphocytes are clearly defective in caveolin (see Figure 8). Indeed, new data from primary fresh human T-cells (Figure 8-10), which do not express caveolin, further support our claims. Similar to Jurkat, Kv1.3 WT and CBDless channels trigger similar results, but when caveolin 1 was introduced, as requested, only in WT-transfected cells, apoptosis was partially counteracted (Figure 10).

• Is increased apoptosis in 3T3-L1 cells following caveolin-1 knockdown due to the pro-apoptotic effects of mitoKv1.3 (as you suggest), or is it due to p53-p21-dependent induction of mitochondrial dysfunction and cellular senescence caused by caveolin-1 deficiency (Aging Cell 2017;16:773-784)? One could distinguish between these possibilities by over-expressing Kv1.3 in 3T3-L1 cells such that there is an excess of mitoKv1.3 over caveolin-1. If apoptosis is enhanced, it would suggest that mitoKv1.3 is a driver of apoptosis.

We thank the reviewer for this suggestion. The reviewer is right that the paper published in Aging cell confers such a role to Cav. On the other hand, deficiency of caveolin itself may trigger mitodysfunction and cellular senescence. In fact, caveolin has been postulated as protective against apoptosis and cell death (just few examples: Yang et al., Cancer Invest. 2012, 30:453462; Codenotti et al., Cancer Lett. 2021, 505:1-12; Aberg et al., Apoptosis. 2020, 25:519534). In this context, we claim that the function of Kv1.3 should be considered within the association to caveolin. All our data support this claim and is concomitant with a partial role for caveolin as mentioned by the reviewer.

Unfortunately, 3T3-L1 cells are extremely resistant to transfection and the experiment suggested is hard to achieve. We tried several times to perform this experiment, but over-expression of Kv1.3 (chemical transfection or electroporation), getting healthy cells to work with, was impossible to achieve in our hands. However, merging the rest of experiments, we may suggest that the axis of interaction between Kv1.3 and caveolin, not just Kv1.3, is at the onset of the apoptotic response. Thus:

1. The overexpression of Kv1.3 CBDless, but not Kv1.3 WT, in HEK-293 and B16F10 melanoma cells, which accumulates in mitochondria, without caveolin association, triggers apoptosis.

2. The overexpression of Kv1.3 WT and CBDless in regular Jurkat (cav -) cells yielded similar apoptosis. However, the use of Jurkat (cav+) cells – see above – indicated that apoptosis was partially prevented solely in Kv1.3 WT, which has the capacity of caveolin association.

3. Similar results were obtained with fresh peripheral human T-lymphocytes. These native cells, as many leukocytes, do not express caveolin (Hatanaka et al., Biochem Biophys Res Commun 1998, 253: 382-387; Vallejo and Hardin, FASEB J. 2005, 19:586-587; Sawada et al., Blood. 2010, 115:2220-2230). The introduction of caveolin protected partially against apoptosis in Kv1.3 WT, but not in Kv1.3 CBDless.

4. In 3T3-L1 cells, caveolin may be ablated by shRNA, thereby an increase in the Kv1.3/caveolin ratio, which surely impairs the Kv1.3-caveolin association, results in more apoptosis. These results parallel T-cell data.

• Human T cells express caveolin-1, and caveolin-1 deficiency in T cells reduces effector function (J Immunol. 2017; 199: 874-884).

Although Borger et al. claim the expression of caveolin 1, caveolin 1 is only observed upon concentration by immunoprecipitation. In contrast, the defective expression of caveolin in lymphocytes has been reported in several papers by different groups (Hatanaka et al., Biochem Biophys Res Commun 1998, 253: 382387; Vallejo and Hardin, FASEB J. 2005, 19:586-587; Sawada et al., Blood. 2010, 115:22202230). It is true that Hatanaka et al., 1998 suggest that certain cell lines may express caveolin under certain states of activation. Our data agrees with that observation (see our new panel in Figure 7D). In fact, we detected, in a regular WB with no immunoprecipitation, a faint band of cav 1 (Figure 7D). Therefore, we mention that cav1 expression is limited. Anyway, Kv1.3 is much more abundant than caveolin. However, as abovementioned, we managed to select some Jurkat cav+ (Figure 7D) and results in this cell clone, as well as primary T-lymphocytes support our claim.

Based on your results, caveolin-1 deficient primary human T cells would be predicted to express less surface Kv1.3 and more mitoKv1.3, exhibit altered mitochondrial morphology and physiology, and be more susceptible to apoptosis. This might be worth testing.

As abovementioned, in deficient Jurkat T-cells (regular cav-) both channels (WT and CBDless) triggered similar level of apoptosis and this is partially counteracted by the endogenous expression of caveolin (Jurkat cav+) (Figure 7E). However, as suggested by the reviewer, we have performed several studies in freshly isolated primary human Tlymphocytes (Figure 8-10). In these T-cells, with no endogenous caveolin, apoptosis was partially counteracted in Kv1.3 WT-expressing cells, but not in CBDless cells, once cav was introduced. In addition, both Kv1.3 channels share similar location in primary T-cells with no caveolin (Figure 8). As mentioned by the reviewer, Pearson’s coefficient of colocalization is higher in mitochondria than membrane. However, this should be taken with caution because membrane and mitochondrial marker staining are quite different.

In addition, new data on primary human T-lymphocytes (Figure 9-11) demonstrate that the reviewer is correct and the expression of Kv1.3 triggers “*altered mitochondrial morphology and physiology, and cells are more susceptible to apoptosis*”. Overall, our data supports that the association Kv1.3/cav is important for fine-tuning the apoptosis susceptibility.

• Accumulation of Kv1.3 in the ER and Golgi when CBD is mutated (Figure 2D) may affect the physiology of these organelles. The authors may wish to address/discuss this.

Thank the reviewer for this suggestion. The increment of colocalization of Kv1.3CBDless with these organelles is a result of a massive increase of intracellular retention. However, Supplemental Figure 7 we show no ER stress. Anyway, in order to address this specific concern, in this new version, we have used a Kv1.3 mutant (YMVIii) that shows massive ER intracellular retention and does not reach the membrane surface (Martinez-Marmol et al., J Cell Sci. 2013, 126:5681-5691). The Kv1.3 YMVIii mutant, associates with caveolin, colocalizes within the ER (Martinez-Marmol et al., 2013), routes to mitochondria, targets no plasma membrane (Martinez-Marmol et al., 2013) and causes no apoptosis (Supplemental Figure 6D). Therefore, data obtained with this channel further support that the Kv1.3/caveolin association rather than the intracellular retention by itself is the main responsible for our claims. These results are introduced now as a Supplemental Figure 6.

Reviewer #3:Capera et al. cover an interesting topic in their manuscript, which reports a novel role of the interaction between caveolin 1 and Kv1.3 channels. The Authors present a detailed analysis on the possible function of caveolin and Kv1.3 in cellular processes such as apoptosis. They revealed that mitochondrion associated Kv1.3 without caveolin 1 binding domain is pro-apoptotic, WT Kv1.3 show colocalization with caveolin in the mitochondrial inner membrane, contributes to the mitochondrial dysfunction via destructing the structure of cisterna. Authors concluded that Kv1.3-caveolin 1 "crosstalk" can sensitize the cancer cells to apoptosis. Based on these my questions and comments are the following:

Thank you very much for your kind report considering that our novel work covers an interesting topic undertaking a detailed study. We apologize for the delay answering the reviewers concerns. The initial and total lockdown in both Spain and Italy plus further partial lockdowns during the last months have made very difficult to address all requests in due time.

1. Authors in a series of experiments use Cav1 knock-down HEK cell line. However, I did not find in the manuscript if they applied it in the mitochondrial function and apoptosis experiments. As all the experiments related to Cav1-driven Kv1.3 targeting I suppose it would be practical to use a cell line lacking endogenous Cav1 and then "re-add" Cav1 with transfection of Cav1 plasmid e.g. for organelle-localization experiments, mitochondrial location, apoptosis.

We only used the HEK cav1- cell line when FRET between cav and Kv1.3 was analyzed (Figure 1D). The rest of the paper deals mostly with HEK-293 wt cells and lymphocytes. In fact, the HEK cav1- was not of interest in this work. This cell line was widely characterized in a previous work (Perez-Verdaguer et al., Sci Rep. 2016, 6:22453). As requested, we have introduced a set of new experiments with primary human T-cells, which have limited expression of cav, and introduced cav in order to evaluate what the Reviewer proposes. Importantly, our new data in primary human T-cells (Figure 8-10) and Jurkat (Figure 7) demonstrate that the presence of cav partially protects from apoptosis. In summary, taking into account all cellular models (HEK, B16F10 melanoma, Jurkat, primary human T-cells and NIH 3T3 L1 cells) allows us achieving relevant conclusions.

2. For electrophysiology Authors chose oocytes instead of HEK cells. Why did not the authors show data which were measured in HEK cells?

The aim of the present work was to decipher whether the association of Kv1.3 and caveolin influences cell survival and apoptosis. We initially wanted to perform the electrophysiology in our mammalian cell models rather than in oocytes. However, as abovementioned, Kv1.3CBDless generates important changes at the membrane physiology and integrity triggering membrane weakness impeding patch clamp experiments.

As mentioned above, CBDless cells undergo severe apoptosis and changes in membrane lipid composition and membrane integrity are evident. Because of that, and patch-clamp being impossible to achieve in mammalian culture cells, we analyzed the cholesterol and sphingolipid concentration of Kv1.3CBDless HEK cells. CBDless cells exhibited a reduction of 12% cholesterol, 10% phosphatidylserine + phosphatidylinositol and 5% sphingolipids in their membranes. This clearly will affect the integrity of the membrane as well as the kinetic of channels (Cholesterol regulation of Ion channels and receptors, I. Levitan and F.J. Barrantes eds. 2012. DOI: 10.1002/9781118342312. John Wiley and Sons, Inc).

Probably because of this reason, seals were extremely instable and cells died just applying the whole cell configuration. To sum up, mammalian cell models were not suitable for electrophysiology.

Our choice of using oocytes was driven by the observation that these cells exhibit an extremely resistant membrane for the validation of channel tetramerization and functionality. In any case, the rationale for the electrophysiological characterization of the CBDless mutant was the fact that altering the CBD, lying next to the tetramerization domain, could impair the oligomerization and functionality of the channel. Therefore, we wanted to exclude the possibility that Kv1.3 CBDless gives rise to a non-functional channel, but the detailed electrophysiological characterization of this mutant was not the primary scope of the current study. In our previous version, we analyzed few parameters to demonstrate that the CBDless was functional in terms of tetramerization and conductivity. The effects of a limited caveolin interaction of the channel on the electrophysiological parameters of the CBDless mutant were addressed previously (Perez-Verdaguer et al., Sci Rep 2016;6:22453).

In their previous paper the authors demonstrated that in Cav1- HEK cells the Kv1.3 channel had slower inactivation kinetics than in those with Cav1 expression, and not a drastic reduction in the current happened upon Cav1 knock-down. Here the data show that prevention of Cav1 binding to Kv1.3 channels results in a drastic change. I think these outcomes, at least in part, are contradictory. Do the Authors have any explanation for that?

The Reviewer raises and interesting point that is currently under study in our laboratory. In our previous study, knock-down of Cav1 still allowed Kv1.3 to reach the plasma membrane, at least partially. Here we show that the same is true for the Kv1.3 CBDless, where there is no interaction at all with Cav1, and thus the intracellular retention of the mutant channel is further enhanced. As mentioned above, to establish a direct correlation to answer the question of the reviewer, the electrophysiological parameters should be studied in HEK293 cells, which is however impossible in the case of the CBD-less Kv1.3.

Is it possible that mutation eliminated the interaction between the channel and another (not Cav1) protein necessary for PM targeting?

The reviewer is correct and because the CBD lies close the tetramerization domain as well as to the Kvβ binding domain, we have checked both possibilities. Throughout the manuscript we show that Kv1.3CBD less mutant tetramerizes (Supplemental Figure 2) being functional (Supplemental Figure 3) and we now include for the internal reviewer’s evaluation some preliminary raw data showing that the CBDless mutant is also able to associate to Kvβ subunits. Although we have addressed the possible elimination of the channel interaction with the auxiliary subunit due to mutation, lying next to the CBD cluster, the possibility of any alternative protein remains open; therefore further studies need to be performed, that are out of the scope of the paper on our opinion. For this reason, we decided to show these data here to the Reviewer, but not to include it into the manuscript. Another important issue could be the interaction of Kv1.3 with lipids. The apoptosis generated by the CBDless mutant trigger an important decrease of membrane cholesterol. The effects of cholesterol on ion channels have been widely documented (Cholesterol on Ion channels, book, Levitan and Barrantes editors).

3. For the TEM images the authors mentioned that Kv1.3 channels are localized to mitochondria detected using the immunogold labeling technique. I could observe only one-two gold beads/mitochondria (or organelles), which I suppose is unexpectedly low as Kv1.3 protein was overexpressed (and also large fraction of channels target to mitochondria). On the other hand, I wonder if any specific markers for organelles were applied to identify them.

The reviewer is right, but please note that in the immunogold TEM, images of a single, tiny section of mitochondria are visible. Figure 7C shows two spots for Kv1.3 and only one spot for Cav, while both proteins are well-detectable in purified mitochondria by Western blot. We have to take into consideration that in the TEM images, using a very limited dilution of antibody, as we did use, we assure that the staining is specific. More staining often represents increase in background, false labeling and poor structural preservation.

[Editors’ note: what follows is the authors’ response to the second round of review.]

The manuscript has been improved but there are some remaining issues that need to be addressed, as outlined below:1. One of the claims made in the paper: " interactions between caveolin and Kv1.3 are essential for apoptosis" seems slightly overstated. We recommend the authors to soften this claim and discuss alternative possibilities. Their work clearly provides evidence for improper trafficking of Kv1.3 CBDless to the plasma membrane and ectopic expression in mitochondria, resulting in apoptosis. They have also shown that caveolin is involved in modulating apoptosis. Previous work from their lab has shown caveolin interacts with Kv1.3 and increases C-type inactivation and fails to interact with CBDlessKV1.3. It remains unclear whether there is any direct interaction/cross-modulation between Kv1.3 and caveolin in the mitochondria, causing apoptosis, and that requires more direct evidence to support the claim.There is no doubt that both caveolin and Kv1.3 are involved in apoptosis and apoptosis IS dependent on Kv1.3-caveolin axis, but the direct interaction between these two proteins in the mitochondria warrants further investigations. The experimental evidence provided in the manuscript are suggestive of such an interaction in mitochondria and resulting apoptosis, but it is not definitive. Addressing it in the discussion will benefit the readers.

We thank the Reviewer for this important comment. Indeed, we agree that direct interaction of mitoKv1.3 with mitochondria-located caveolin is not proven in our experiments. Thus, as kindly suggested, our claims have been soften thoroughly the manuscript.

We have modified the Discussion accordingly and a new sentence has been included (p 11, l 16-19): “Although a direct interaction of these proteins in mitochondria has not been confirmed and warrants further investigation, evidence suggests that apoptosis is dependent on mitoKv1.3-caveolin functional axis”.

In addition, we have deleted the following sentence from the Discussion: “The severity of the Kv1.3-mediated events highly depends on its intramitochondrial association with Cav1.”

We have also modified the subheading (p 7) and a sentence in the Results section: “Our data indicated that altering the functional crosstalk between Cav1 and Kv1.3 plays a crucial role in determining the sensitivity of cells to apoptosis” (p 8, l 21-22). This statement emphasizes that interaction between mitoKv1.3 and mitochondria-located Cav1 is functional (not necessarily physical). To note that, functional crosstalk is supported by several observations reported in our work, including that, unlike Kv1.3 CBDless, Kv1.3 (YMVIii) partially locates to mitochondria, associates to Cav 1 (Suppl Figure 6C), but does not trigger relevant apoptosis (Suppl Figure 6D).

2. Regarding the electrophysiological properties of the mutant channel, against the view of the authors and even though the construct appears toxic to mammalian cells and the experiments need to be done in oocytes, if the mutant activates „earlier", it is in our view more active at any potential. Also, in oocytes, if the transport to the membrane is altered, then it would be expected that the currents are smaller. A decrease in the input resistance of the oocyte would also rather indicate a "higher" basal activity. If we understand correctly, the role of Kv1.3 in apoptosis relies on the fact that reduction of its activity through bax blockade triggers ∆ψm hyperpolarization and then apoptosis, so it is counterintuitive how a "more active" channel is proapoptotic.

We would like to emphasize that currents generated by the CBDless mutant should be taken with caution and mostly considered artefacts because the massive fail of cellular integrity. The electrophysiological characterization of Kv1.3 CBDless was undertaken exclusively to demonstrate that the mutant channel was functional because the CBD lies near the tetramerization domain. CBDless mutant currents are deeply affected by the massive cellular apoptosis, which affects the integrity of their plasma membrane and lipid composition (see Cholesterol Regulation of Ion Channels and Receptors, Levitan and Barrantes eds. John Wiley and Sons Inc; 2012).

Anyway, the reviewer is correct and the conductance/voltage curve of the Kv1.3 CBDless shifts to left indicating that the channel activates a more negative potentials (earlier activation). However, because the membrane expression is very different the K^+^ conductance at -40 mV (close to the resting membrane potential of oocytes) is similar for CBDless and WT (5.89 ± 1.20 μS, n=28, vs 5.93 ± 0.98 μS, n=23, for CBD-less and WT, respectively) what explains why the membrane potential of both groups was similar. These values cannot be correlated with input resistance because the input resistance is measured at -100/-80 mV and Kv1.3 is not active at the plasma membrane. Therefore, this value only indicates passive properties of oocyte membrane. The fact that the CBDless mutant exhibited a less input resistance is because the membrane is more leaky that WT because cells are sick (apoptosis).

Importantly, how activation of Kv1.3 happens with the values of ∆ψm in the context of mitochondria (140 mV negative to the activation potential of Kv1.3) remains an unanswered question. We think that this issue merits a few words in the discussion. Moreover, Cav1 maintains mitochondrial depolarization, as its knockdown induces hyperpolarization. This is against what one would expect if it acts partly through Kv1.3. The concept "Kv1.3-mediated cell death" is somewhat misleading if Kv1.3 blockade is what triggers apoptosis. Why does then mitoKv1.3 overexpression promote apoptosis? We might well have got the point wrong, but we think other readers would have the same problem.

Many thanks for calling our attention to the need of explaining better the somewhat complex physiology of mitochondrial channels. The Reviewer is correct that unfortunately it is still unknown how mitoKv1.3 can be active at the highly negative membrane potential found across the inner mitochondrial membrane (IMM) (around -180mV). As mentioned in the Introduction, what was shown is that acute inhibition of mitoKv1.3 causes IMM hyperpolarization, and this triggers ROS production with subsequent opening of the permeability transition pore (PTP). When PTP opens, this leads to loss of mitochondrial integrity, IMM depolarization, swelling and cytochrome c release (now we added this sentence to accentuate the consequences of PTP opening). PTP opening can be triggered, among other factors, either by ROS (triggered by hyperpolarization due to the chemical reduction of respiratory chain complexes) or by IMM depolarization above a certain threshold. Overexpression of K^+^ transporting pathways in mitochondria causes depolarization, as they allow the influx of depolarizing K^+^ into the matrix following the electrochemical gradient for this ion and cause mitochondrial depolarization as well as changes in ultrastructure (see e.g. Figure 3d, Paggio et al., Nature. 2019;572(7771):609-613.). Thus, acute changes in the membrane potential, upon block of the channel, induce ROS and PTP opening, while overexpression of mito-located Kv1.3, that is more prominent in the case of CBDless mutant or in the absence of Cav1 (for both WT and CBD-less Kv1.3), causes sustained depolarization that sensitizes the cells to apoptotic stimuli (Suppl Figure 5). PTP opening then further depolarizes IMM and reduces respiration due to swelling and loss of cytochrome c from the respiratory chain. Overexpression of WT or CBDless equally depolarizes mitochondria and causes apoptosis in primary T cells that lack endogenous Cav1. It has to be mentioned that while overexpression promotes apoptosis, the lack of the channel (WT Kv1.3) renders the cells resistant to apoptosis (Szabo et al., PNAS, 2008). This situation is similar to other ones, where the pharmacological block of a channel (e.g. KATP; Miki et al., Proc Natl Acad Sci U S A 1998;95(18):10402-6) does not yield the same effect of KO, as the former one triggers a series of signaling events that cannot be triggered in the cells lacking the channel. We now introduced the above considerations in the Introduction (page 4, lines 18-19) and Discussion (p 9, l 18-34) as suggested.

In addition, the role of Cav1 in mitochondrial function is under debate. While the reviewer mentions: *“Cav1 maintains mitochondrial depolarization, as its knockdown induces hyperpolarization”*, Yu et al., (Aging Cell. 2017; 16(4):773-784) claim that Cav-1 knockdown prevents mitochondrial respiration and ATP production. In fact, the enzymatic activity of OXPHOS complex I after Cav-1 knockdown was reduced to approximately 40%. This fact would trigger depolarization rather hyperpolarization. Furthermore, we have recently shown that mitoKv1.3 interacts with complex I and inhibition of mitoKv1.3 reduces complex I activity (Peruzzo et al., 2020 Redox Biology; 37:101705). Therefore, it is tempting to speculate that if Cav1 knockdown reduces complex I activity, and mitoKv1.3 blockage reduces complex I activity, a functional link between Cav1 and mitoKv1.3 would exist at the level of complex I further supporting our claim. However, this is far beyond the scope of the present manuscript.

3. Disregarding the fact that Kv1.3 overexpression on its own reduces respiration by almost one-half (Figure 5), the mitochondrial fragmentation induced by Kv1.3CBDless deserves some comments in this context.

Basal (without addition of oligomycin) (Figure 5B) and maximal respiration (Figure 5C) were slightly reduced (by 15%) in the case of WT Kv1.3, while reduction was significantly higher for Kv1.3 CBD-less. The cells were transfected directly in the Seahorse plate and efficiency (more than 60% of the cells were transfected and transfection efficiency was controlled by YFP fluorescence, as mentioned in the Materials and methods section). In any case, in Figure 5C data were normalized with respect to the basal respiration and thus show a significant decrease in maximal respiration independently of the cell number. In general, mitochondrial fission is regulated in a way to couple mitochondrial morphology to the energetic status of the cell (see e.g. Giacomello et al., 2020, Nature reviews Molecular cell biology 21, 204-224). Fission is induced by phosphorylation mediated by AMPK, able to sense the changes in the energy status of a cell and consequently adapt mitochondrial function and dynamics. AMPK facilitates mitochondrial fission downstream of mitochondrial dysfunction caused by mitochondrial respiration inhibition. This mechanism was proposed to facilitate apoptosis in cells with severely dysfunctional mitochondria (Toyama et al., 2016, Science (New York, NY) 351, 275-281). The above explanation has now been added to the Discussion section (p 10, l 30-36).